# Netrin signaling mediates survival of dormant epithelial ovarian cancer cells

Pirunthan Perampalam[1,2], James I MacDonald[1,3], Komila Zakirova[1,3], Daniel T Passos[1,3], Sumaiyah Wasif[1,3], Yudith Ramos-Valdes[1,4], Maeva Hervieu[5], Patrick Mehlen[5,6], Rob Rottapel[7,8], Benjamin Gibert[5], Rohann JM Correa[1,9], Trevor G Shepherd[1,4,9,10,11], Frederick A Dick[1,3,9,12]*

[1]London Regional Cancer Program, London Health Sciences Centre Research Institute, London, Canada; [2]Department of Biochemistry, University of Western Ontario, London, Canada; [3]Department of Pathology and Laboratory Medicine, University of Western Ontario, London, Canada; [4]The Mary and John Knight Translational Ovarian Cancer Research Unit, London Regional Cancer Program, London, Canada; [5]Apoptosis, Cancer and Development Laboratory - Equipe labellisée 'La Ligue', LabEx DEVweCAN, Institut Convergence PLAsCAN, Centre de Recherche en Cancérologie de Lyon (CRCL), INSERM U1052-CNRS UMR5286, Université de Lyon, Université Claude Bernard Lyon1, Centre Léon Bérard, Lyon, France; [6]Netris Pharma, Lyon, France; [7]Princess Margaret Cancer Centre, University Health Network, Toronto, Canada; [8]Department of Medical Biophysics, University of Toronto, 1 King's College Circle, Toronto, Canada; [9]Department of Oncology, Western University, London, Canada; [10]Department of Obstetrics and Gynecology, Western University, London, Canada; [11]Department of Anatomy and Cell Biology, Western University, London, Canada; [12]Children's Health Research Institute, London, Canada

*For correspondence:
fdick@uwo.ca

**Abstract** Dormancy in cancer is a clinical state in which residual disease remains undetectable for a prolonged duration. At a cellular level, rare cancer cells cease proliferation and survive chemotherapy and disseminate disease. We created a suspension culture model of high-grade serous ovarian cancer (HGSOC) dormancy and devised a novel CRISPR screening approach to identify survival genes in this context. In combination with RNA-seq, we discovered the Netrin signaling pathway as critical to dormant HGSOC cell survival. We demonstrate that Netrin-1, –3, and its receptors are essential for low level ERK activation to promote survival, and that Netrin activation of ERK is unable to induce proliferation. Deletion of all UNC5 family receptors blocks Netrin signaling in HGSOC cells and compromises viability during the dormancy step of dissemination in xenograft assays. Furthermore, we demonstrate that Netrin-1 and –3 overexpression in HGSOC correlates with poor outcome. Specifically, our experiments reveal that Netrin overexpression elevates cell survival in dormant culture conditions and contributes to greater spread of disease in a xenograft model of abdominal dissemination. This study highlights Netrin signaling as a key mediator HGSOC cancer cell dormancy and metastasis.

## eLife assessment

The authors further corroborated their model that Netrin signaling promotes survival and dissemination of non-proliferating ovarian cancer cells. These **valuable** results were found to be of significant potential interest to cancer biologists in as much as they address gaps in knowledge pertinent to the mechanisms underpinning ovarian cancer spread. In general, it was thought that

**solid** experimental evidence was provided to support the role of Netrin signaling in fueling ovarian cancer progression.

## Introduction

Relapse from initially effective cancer treatment represents one of the greatest barriers to improving outcomes (*Giancotti, 2013*). Dormant or minimal residual disease that is not clinically detectable is a common source of relapse, particularly in a metastatic setting (*Massagué and Ganesh, 2021*). This has motivated the search for, and study of, disseminated cancer cells. Cellular dormancy has been characterized in cancer cells derived from a variety of primary disease sites including breast, prostate, melanoma, and others (*Sosa et al., 2014*; *Phan and Croucher, 2020*). Disseminated cells often occupy specific cellular niches in perivascular space or in bone marrow (*Goddard et al., 2018*). They adopt catabolic metabolism (*Massagué and Ganesh, 2021*), and withdraw from the cell cycle (*Phan and Croucher, 2020*). Stem like phenotypes and epithelial to mesenchymal transition also characterize the biology of these cells as they survive dissemination and lay quiescent in distant tissues (*Giancotti, 2013*; *Sosa et al., 2014*; *Goddard et al., 2018*). Across a variety of disease sites this dormant state is characterized by diminished Ras-MAPK signaling and elevated p38 activity (*Sosa et al., 2014*; *Yeh and Ramaswamy, 2015*). To develop treatments specifically tailored for dormant cells, a systematic search for vulnerabilities unique to dormancy is needed. To this end, high-grade serous ovarian cancer (HGSOC) disseminates in ascites as growth arrested cellular aggregates (*Correa et al., 2012*). This has been modeled in culture (*Lengyel et al., 2014*), and studies confirm active induction of cell cycle arrest mechanisms (*Hu et al., 2011*; *MacDonald et al., 2017*), induction of autophagy (*Correa et al., 2014*; *Correa et al., 2015*), and other known characteristics of dormancy (*Shepherd and Dick, 2022*). This suggests that HGSOC spheroids are ideal to search for survival dependencies specific to dormant cells.

HGSOC is the deadliest gynecologic malignancy, accounting for more than 70% of cases (*Lheureux et al., 2019*). Its poor prognosis is largely attributed to late-stage diagnosis with metastases already present. Metastatic spread is facilitated by multicellular clusters of tumor cells, called spheroids, that are shed into the peritoneal cavity and disseminate to nearby organs (*Lengyel, 2010*). The non-proliferative state of spheroid cells renders them chemoresistant and contributes to disease recurrence, emphasizing the need to understand the biology of these cells (*Bowtell et al., 2015*). Dormant HGSOC cells are characterized by similar metabolic changes, EMT, and cytokine signaling as dormant cancer cells from other disease sites (*Sosa et al., 2014*; *Shepherd and Dick, 2022*). Mechanisms that mediate their dormant properties include AMPK-LKB1 signaling that induces autophagy (*Peart et al., 2015*; *Buensuceso et al., 2020*), STAT3 and FAK for survival and chemoresistance (*Chen et al., 2017b*; *Diaz Osterman et al., 2019*), and DYRK1A signaling to arrest proliferation (*MacDonald et al., 2017*). Unfortunately, therapeutic approaches to target these pathways in ovarian cancer have yet to emerge, indicating that more candidates are needed.

Netrins are a family of secreted factors that reside in the extracellular matrix and were first identified in axon guidance in the developing nervous system (*Rajasekharan and Kennedy, 2009*; *Kryza et al., 2023*). Together with their receptors, Netrins have since been implicated in cancer progression (*Mehlen et al., 2011*). Netrin-1 is the most studied and its misregulation in cancer leads to pro-survival signals (*Arakawa, 2004*). Netrin interactions with DCC and UNC5 homologs (UNC5H, UNC5A-D) convert pro-death signals by these dependence receptors to survival signals (*Brisset et al., 2021*). Netrin signals through PI3K-AKT and ERK-MAPK are known to participate in orientation decisions in development by providing guidance cues (*Forcet et al., 2002*; *Larrieu-Lahargue et al., 2010*; *Yin et al., 2017*), while loss of ligand binding leads DCC or UNC5H to stimulate dephosphorylation of DAPK1 that sends pro-death signals (*Llambi et al., 2005*; *Castets et al., 2009*). The gene encoding DCC is frequently deleted in cancer (*Arakawa, 2004*), and Netrin-1 blocking antibodies are in Phase II clinical trials for endometrial and cervical carcinomas to inhibit metastatic growth (*Brisset et al., 2021*). Alternative receptors for Netrins such as NEO, DSCAM, and Integrins also mediate their signals (*Mehlen et al., 2011*), creating a myriad of possible pathways for Netrins to regulate cancer cells. Lastly, functional roles for Netrin-3,–4, and –5 are still relatively unexplored compared to Netrin-1 (*Bruikman et al., 2019*), although their expression patterns suggest roles in advanced forms

**eLife digest** High-grade serous ovarian cancer (or HGSOC for short) is the fifth leading cause of cancer-related deaths in women. It is generally diagnosed at an advanced stage of disease when the cancer has already spread to other parts of the body. Surgical removal of tumors and subsequent treatment with chemotherapy often reduces the signs and symptoms of the disease for a time but some cancer cells tend to survive so that patients eventually relapse.

The HGSOC cells typically spread from the ovaries by moving through the liquid surrounding organs in the abdomen. The cells clump together and enter an inactive state known as dormancy that allows them to survive chemotherapy and low-nutrient conditions. Understanding how to develop new drug therapies that target dormant cancer cells is thought to be an important step in prolonging the life of HGSOC patients.

Cancer cells are hardwired to multiply and grow, so Perampalam et al. reasoned that becoming dormant poses challenges for HGSOC cells, which may create unique vulnerabilities not shared by proliferating cancer cells. To find out more, the researchers used HGSOC cells that had been isolated from patients and grown in the laboratory. The team used a gene editing technique to screen HGSOC cells for genes required by the cells to survive when they are dormant.

The experiments found that genes involved in a cell signaling pathway, known as Netrin signaling, were critical for the cells to survive. Previous studies have shown that Netrin signaling helps the nervous system form in embryos and inhibits a program of controlled cell death in some cancers. Perampalam et al. discovered that Netrins were present in the environment immediately surrounding dormant HGSOC cells. Human HGSOC patients with higher levels of Netrin gene expression had poorer prognoses than patients with lower levels of Netrin gene expression. Further experiments demonstrated that Netrins help dormant HGSOC cells to spread around the body.

These findings suggest that Netrin signalling may provide useful targets for future drug therapies against dormant cells in some ovarian cancers. This could include repurposing drugs already in development or creating new inhibitors of this pathway.

of some cancers (*Jiang et al., 2021*). A functional role for Netrins and their receptors in dormancy, or other aspects of HGSOC spheroid biology have yet to be reported.

In this study, we utilized a suspension culture system to interrogate survival mechanisms in dormancy using HGSOC spheroids. To maximize discovery of gene loss events that compromise viability in dormancy, we devised a novel genome-wide CRISPR screen approach that we term 'GO-CRISPR'. In addition to the standard CRISPR workflow, GO-CRISPR utilizes a parallel screen in which 'guide-only' cells lacking Cas9 are used to control for stochastic changes in gRNA abundance in non-proliferative cells. This screen identified multiple Netrin ligands, DCC, UNC5Hs, and downstream MAPK signaling components as supporting survival of HGSOC cells in dormant culture conditions. A transcriptomic analysis of HGSOC spheroids also revealed Netrin signaling components were enriched in HGSOC spheroids compared to adherent cells, and that in the absence of the DYRK1A survival kinase Netrin transcriptional increases were lost. We show that multiple Netrins, UNC5 receptors, and the downstream MEK-ERK axis is crucial for survival signaling in dormant cell culture. Importantly, Netrin activation of ERK is restricted to spheroid cultures and is unable to stimulate proliferation, consistent with its role in dormant survival. We blocked Netrin signaling by deleting UNC5 receptors and using xenograft approaches demonstrated that loss of Netrin signaling compromises survival of HGSOC cells, specifically in the dormant stage of dissemination. Furthermore, we show that overexpression of Netrin-1 and –3 are associated with poor prognosis in HGSOC and their overexpression in culture increases survival in suspension culture. Netrin overexpression in xenografts contributes to increased disease dissemination in a model of metastasis. Our study highlights Netrin-ERK signaling as a requirement for spheroid survival and suggests it may be a potential therapeutic target for eradicating dormant HGSOC cells.

## Results

### Axon guidance genes are essential factors for HGSOC spheroid viability

To dissect genetic dependencies of dormant ovarian cancer cells, we utilized a suspension culture system that encourages cellular aggregation, leading to non-proliferative spheroids that resemble dormant aggregates found in ovarian cancer patient ascites (*Correa et al., 2012*; *MacDonald et al., 2017*; *Shepherd and Dick, 2022*; *Shepherd et al., 2006*). *Figure 1A* illustrates withdrawal from cellular proliferation by HGSOC cells in suspension as evidenced by reduced expression of Ki67 and increased expression of the quiescence marker p130 (*Litovchick et al., 2004*). Furthermore, phospho-western blots for ERK demonstrate reduced but not absent phosphorylation, and phospho-p38 either remains consistent or is increased. Reduced phospho-ERK and elevated phospho-p38 are indicative of the characteristic signaling switch reported to underlie most paradigms of cancer cell dormancy (*Goddard et al., 2018*). These findings are relatively consistent among the three HGSOC cell lines; iOvCa147, OVCAR8, TOV1946 selected for use in our screen. To discover genes required for survival in spheroids, we utilized a new approach called GO-CRISPR (Guide Only CRISPR) that we developed specifically for survival gene discovery in three-dimensional spheroid cell culture conditions (*Figure 1B*). GO-CRISPR incorporates sgRNA abundances from non-Cas9-expressing cells to control for stochastic (non-Cas9 dependent) effects that are a challenge in three-dimensional culture conditions with slow or arrested proliferation (*Zanoni et al., 2016*). We used sequencing to determine sgRNA identity and abundance in the library, initially infected cells, and in cells following 48 hr of suspension culture (*Figure 1B*). Read count data was analyzed using a software package that we developed to analyze GO-CRISPR data (summarized in *Figure 1*; *Figure 1—figure supplement 1A and B*). We selected genes that had a Spheroid/Adherent Enrichment Ratio (ER) less than 1 (p<0.05) because some genes become depleted in adherent growth before being transferred to suspension culture conditions and this ratio selects for genes with a stronger effect in suspension when cells are not proliferating (*Figure 1B*). To assess reliability of screen data, we investigated the classification of non-targeting control sgRNAs. We computed a receiver operator characteristic curve (ROC) and determined the area under the curve for each cell line (*Figure 1*; *Figure 1—figure supplement 1C*). Greater than 95% of sgRNAs in iOVCA147 and OVCAR8 screens were classified as non-essential, while 83% of sgRNAs in TOV1946 cells were non-essential. In addition, we compared GO-CRISPR screen ER values with viability data from siRNA knock down for 35 individual genes we have previously studied in spheroid survival assays in OVCAR8 and iOVCA147 cells (*Figure 1*; *Figure 1—figure supplement 1D*; *MacDonald et al., 2017*). This revealed a statistically significant correlation between the two sets of data, suggesting that genes we previously knew supported spheroid viability (or were dispensable) were classified similarly by our screen. These controls suggested our screen data was reliable for identifying new survival dependencies.

We found 6,717 genes with an ER below 1 in iOvCa147 cells; and 7,637 genes in TOV1946; and 7,640 genes in OVCAR8 cells. Among these genes; 1382 were common to all three cell lines (*Figure 1D*). We performed pathway enrichment analysis on these common genes and identified several significantly enriched pathways. These include primary and secondary Reactome gene sets such as signal transduction, metabolism, and signaling by GPCR (*Figure 1E*). In addition, we identified a tertiary category called 'axon guidance,' that includes numerous components of the Netrin signaling pathway that scored highly.

### Expression of Netrin ligands and receptors are enriched in HGSOC spheroids

DYRK1A was previously discovered to support survival of ovarian cancer cells in suspension culture (*MacDonald et al., 2017*). Since DYRK1A phosphorylates RNA polymerase II in transcription initiation (*Di Vona et al., 2015*), we sought to determine if it regulates gene expression in spheroid survival. We created *DYRK1A⁻ᐟ⁻* iOvCa147 cells and confirmed their deficiency for survival in suspension (*Figure 2*; *Figure 2—figure supplement 1A–E*). Using iOvCa147 and *DYRK1A⁻ᐟ⁻* derivatives, we performed RNA-seq on adherent and spheroid cells. We collected spheroids following a 6 hr incubation in suspension because overexpression of known cell cycle transcriptional targets due to DYRK1A loss were first evident at this time point, but loss of viability due to DYRK1A deficiency had not yet

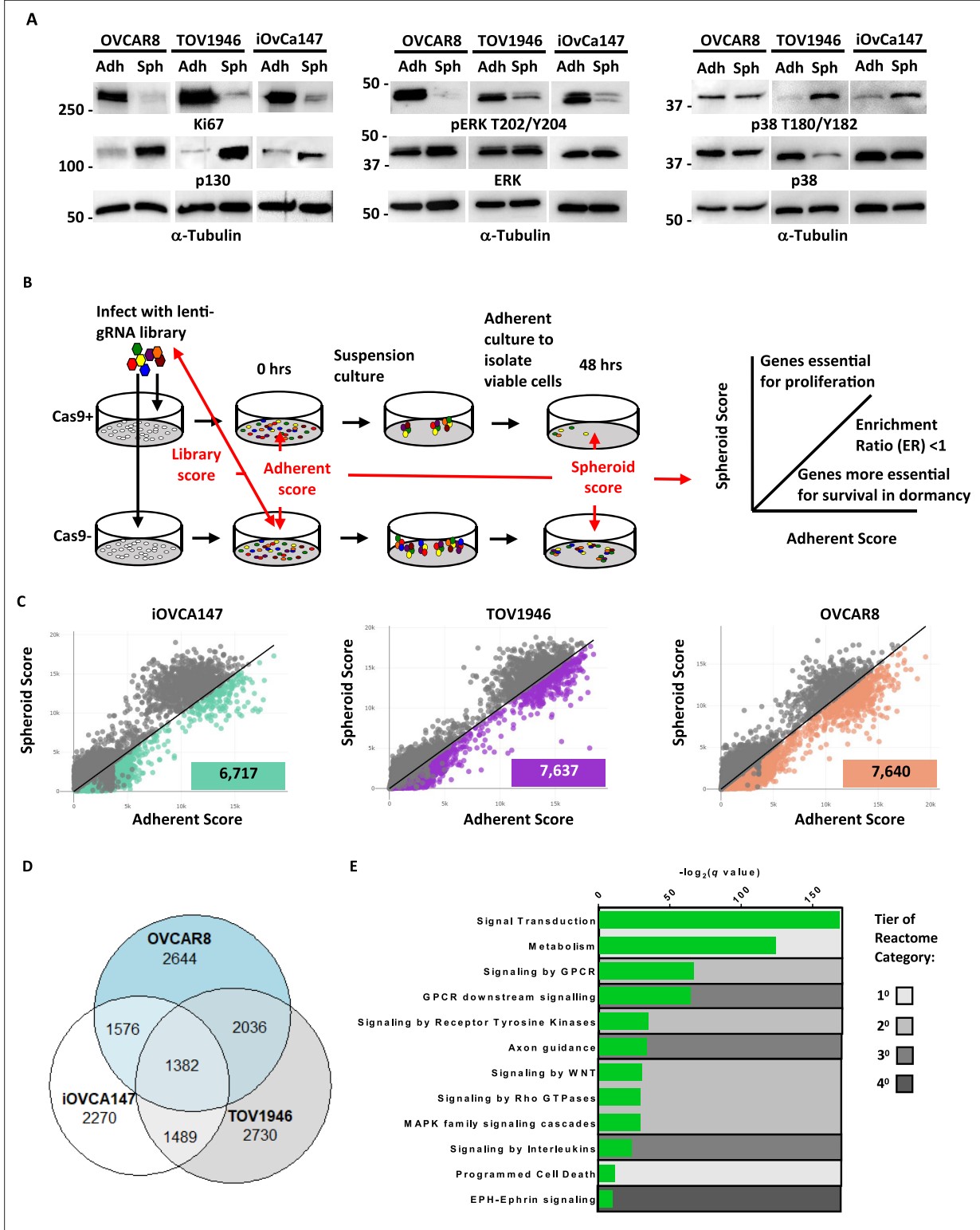

**Figure 1.** GO-CRISPR screens implicate axon guidance pathways as supporting HGSOC spheroid cell viability. (**A**) iOvCa147, TOV1946, or OVCAR8 cells were cultured under adherent conditions (Adh) or in suspension to induce spheroid formation (Sph). Lysates were prepared and analyzed by western blotting for the proliferation marker Ki67 and the quiescence marker p130, and phosphorylated and total levels of ERK and p38. Tubulin was blotted as a loading control. (**B**) Flow chart of GO-CRISPR screening used for each of iOvCa147, TOV1946, or OVCAR8 control cells and derivatives of each that express Cas9. Cas9-positive cells (top row) and Cas9-negative cells (bottom row) were transduced with the GeCKO v2 pooled sgRNA library. After antibiotic selection, cells were expanded under adherent culture conditions (0 hr) before being transferred to suspension culture conditions to induce

*Figure 1 continued on next page*

*Figure 1 continued*

spheroid formation and select for cell survival. After 48 hr, spheroids were transferred to standard plasticware to isolate viable cells. Red arrows indicate the relevant comparisons of sgRNA sequence abundance that were made to analyze screen outcomes. Genes with relatively greater effect on viability in suspension were selected by comparing their scores between adherent and suspension conditions and considering genes with an enrichment ratio (ER) of <1. (**C**) Scatter plots representing the spheroid score on y-axis and adherent score on the x-axis calculated by TRACS for each gene in each cell line (iOvCa147, TOV1946, OVCAR8). Colored data points represent genes with ER <1 and $p_{adj}$ <0.05 (paired t-test). (**D**) Venn diagram illustrating overlap of genes identified as supporting cell viability in suspension culture from iOvCa147, TOV1946, and OVCAR8 cells. (**E**) Graph depicting enriched pathways from ConsensusPathDB using the 1382 commonly identified genes from D. Categories are ranked by q-value. Tiers of Reactome categories are indicated by shading.

The online version of this article includes the following source data and figure supplement(s) for figure 1:

**Source data 1.** Original files for western blot analysis in *Figure 1A* (Ki67, p130, pERK T202/Y204, ERK, p38 T180/Y182, p38, Tubulin).

**Source data 2.** PDF containing annotation of original western blots in *Figure 1A* (Ki67, p130, pERK T202/Y204, ERK, p38 T180/Y182, p38, Tubulin).

**Figure supplement 1.** CRISPR screen analysis details and internal controls.

**Figure supplement 1—source data 1.** Numerical data used in the graph in *Figure 1—figure supplement 1D*.

occurred (*MacDonald et al., 2017*). We compared iOvCa147 adherent cells to 6 hr suspension culture to identify transcriptional changes that occur as these cells transitioned from adherent conditions to suspension conditions (*Figure 2A*). We identified 1834 genes that were upregulated, and pathway enrichment analysis identified many of the same categories as our CRISPR screen including axon guidance (*Figure 2B*). We then compared iOvCa147 spheroid cells to *DYRK1A*<sup>-/-</sup> spheroid cells to identify mRNA expression changes that were lost through DYRK1A deficiency (*Figure 2C*). We identified 744 genes that were downregulated and these also belonged to many of the same pathways, including axon guidance (*Figure 2D*).

We compared enriched pathways from these two RNA-seq analyses with those that were identified in our GO-CRISPR screens to identify gene expression programs in spheroids whose products are essential for cell survival. 78 Reactome pathways were commonly enriched across the GO-CRISPR screens and transcriptomic analyses, including 78 of the 83 pathways identified in RNA-seq from *DYRK1A*<sup>-/-</sup> spheroid cells (*Figure 2E*). The axon guidance pathway was one of the most enriched among these common pathways (*Figure 2F*). These data reveal that our GO-CRISPR screen and transcriptomic analyses converge on this highly specific tertiary Reactome category of axon guidance alongside much broader primary or secondary gene sets. This category includes Netrin ligands, receptors, and downstream signaling components, suggesting a previously unappreciated role in HGSOC spheroid viability.

We separately sought to confirm elevated expression of Netrins and some of their signaling components in HGSOC spheroid cell culture. We examined expression levels of ligands and receptors in the three cell lines used in our GO-CRISPR screen using RT- qPCR (*Figure 3A–C*). The precise identities of upregulated genes varies between cell lines; however, each cell line increases expression of some Netrins, at least one UNC5, and at least one of DCC, DSCAM, and Neogenin. This suggests that the Netrin pathway components are expressed in HGSOC cells may play a previously unappreciated role in dormancy and disease progression.

Since Netrin signaling has not been previously associated with dormancy we examined expression of Netrin-1 and –3 ligands in spheroids isolated directly from patient ascites by immunohistochemistry. First, we assessed proliferative markers in HGSOC tumor tissue and spheroids by staining for Ki67 and p130. As expected, solid HGSOC tumors display uniform Ki67 positivity and relative absence of p130 (*Figure 4A–B*). Conversely, serial sections of spheroids demonstrated rare Ki67 nuclear staining and abundant and uniform p130 staining. This indicates that cells in these patient derived spheroids are quiescent (*Figure 4C–E*). To investigate the presence of Netrins, we initially tested Netrin-1 and –3 antibody specificity using spheroids generated from OVCAR8 cells bearing knock out or over expression of Netrins (*Figure 4*; *Figure 4—figure supplement 1A–F*). Dormant patient derived spheroids were stained for Netrins and epithelial markers Cytokeratin and EpCAM. Since HGSOC is characterized by point mutations in p53 and its accumulation in the nucleus, sections were also stained for p53 to confirm cancer cell identity within these spheroids by high level nuclear staining (*Figure 4F–H*). Overall, this confirms Netrin expression occurs in dormant HGSOC patient spheroids, precisely the context our screen indicates that they are critical for cell survival.

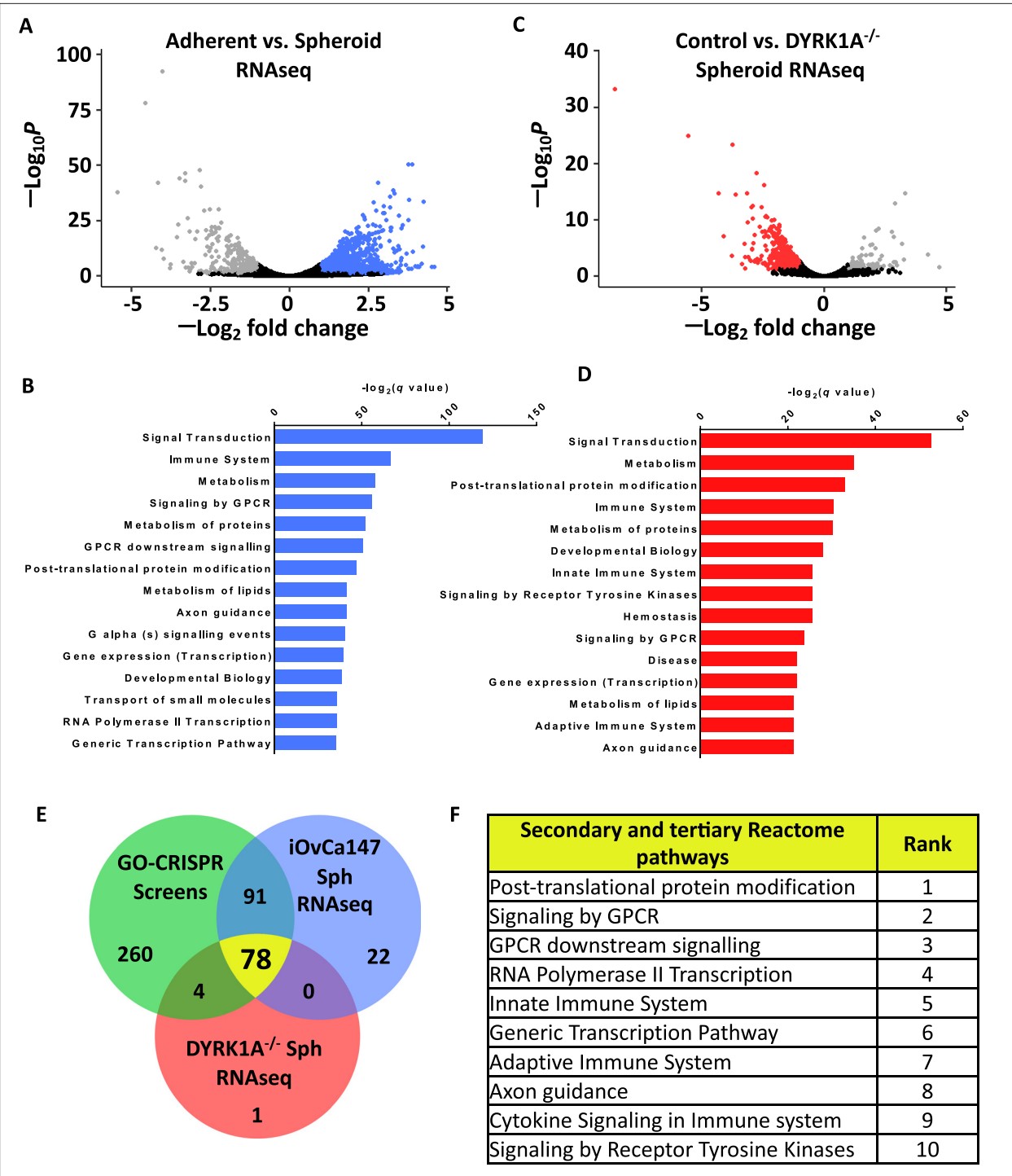

**Figure 2.** Axon guidance pathway components are upregulated in iOvCa147 spheroid cells in a DYRK1A dependent manner. (**A**) RNA was isolated from iOvCa147 cells following culture under adherent conditions or in suspension conditions to induce spheroid formation for 6 hr. Triplicate independent cultures were processed for RNA-seq. A volcano plot shows differentially expressed genes in spheroid cells compared to adherent. 1937 genes were found to be downregulated in iOvCa147 spheroid cells ($\log_2$ fold change <1, $p_{adj}$ <0.05 (Wald test), FDR 10%, highlighted in grey) and 1,834 genes were upregulated ($\log_2$ fold change >1, $p_{adj}$ <0.05, FDR 10%, highlighted in blue). (**B**) Top 15 most significantly enriched pathways ($p_{adj}$ <0.05) whose genes were upregulated in suspension culture compared to adherent in RNA-seq analysis. (**C**) A volcano plot showing differentially expressed genes in *DYRK1A$^{-/-}$* spheroid cells compared to iOvCa147 spheroid cells. A ttoal of 744 genes were found to be downregulated in *DYRK1A$^{-/-}$* spheroid cells ($\log_2$ fold change <1, $p_{adj}$ <0.05 (Wald test), FDR 10%, highlighted in red) and 96 genes were upregulated ($\log_2$ fold change >1, $p_{adj}$ <0.05 (Wald test), FDR 10%, highlighted in grey). (**D**) Top 15 most significantly enriched pathways that were represented by downregulated genes in *DYRK1A$^{-/-}$* suspension

*Figure 2 continued on next page*

*Figure 2 continued*

culture compared to control cells in suspension. (**E**) Venn diagram depicting overlapping enriched pathways identified in GO-CRISPR screens in green; enriched pathways identified in upregulated genes in parental iOvCa147 spheroid cells in blue; and enriched pathways identified in downregulated genes in *DYRK1A*⁻/⁻ spheroid cells in red. 78 pathways were commonly enriched in all three datasets (shown in yellow). (**F**) Top 10 most significantly enriched pathways among the 78 identified in C.

The online version of this article includes the following source data and figure supplement(s) for figure 2:

**Figure supplement 1.** Generation of iOvCa147 cells deficient for DYRK1A and loss of viability in suspension.

**Figure supplement 1—source data 1.** Original files for western blot analysis in *Figure 2—figure supplement 1C and D* – (DYRK1A, pTAU S404, TAU, Tubulin).

**Figure supplement 1—source data 2.** PDF containing annotation of original western blots in *Figure 2—figure supplement 1C and D* – (DYRK1A, pTAU S404, TAU, Tubulin).

**Figure supplement 1—source data 3.** Numerical data used in the graph in *Figure 2—figure supplement 1E*.

## Netrin signaling stimulates ERK to support HGSOC spheroid survival

Netrin ligands and their dependence receptors are best known for their roles in axon guidance (*Meijers et al., 2020*), but are appreciated to regulate apoptosis in cancer (*Mehlen et al., 2011*). *Figure 5A* illustrates components of Netrin signaling and the frequency with which they were discovered to have an ER of <1 in our screens. In addition, intracellular signaling molecules known to be downstream of Netrin receptors that are shared with other enriched pathways identified in our screens, are also illustrated. To confirm screen findings, we independently knocked out *NTN1*, *NTN3*, *NTN4*, *NTN5*, the UNC5H receptor homologs (*UNC5A*, *UNC5B*, *UNC5C*, and *UNC5D*), as well as *DCC*, *DSCAM*, and *NEO1* alone or in multigene combinations using lentiviral delivery of sgRNAs and Cas9 (*Figure 5*; *Figure 5—figure supplement 1A–C*). We assessed the effect of their loss on spheroid survival in six HGSOC cell lines (iOvCa147, TOV1946, OVCAR3, OVCAR4, OVCAR8, and COV318). Loss of at least one Netrin ligand resulted in reduced spheroid cell survival in five of six cell lines (*Figure 5B*). Notably, Netrin-3 loss showed the most pronounced effect on survival. Cell lines such as OVCAR8 and COV318 were relatively resistant to the deletion of Netrin pathway components as individual UNC5, *DCC*, and *DSCAM* genes rarely affected viability (*Figure 5B*). In these instances, combined deletions of UNC5 family members, or *DCC;DSCAM;NEO1* (DDN for short) were either inviable or highly sensitive to suspension culture induced cell death, suggesting redundancy in this pathway in some cell lines. Other cell lines, such as OVCAR3, were highly sensitive to the loss of multiple individual ligands and any individual UNC5H receptor was highly sensitizing to death induction in suspension (*Figure 5B*). Overall, these illustrate a robust role for Netrins in survival in this dormant cell culture assay. Importantly, loss of receptor expression does not elevate viability as expected for a dependence receptors, nor does it induce DAPK1 dephosphorylation (*Figure 5*; *Figure 5—figure supplement 1D*), suggesting that Netrin receptors in HGSOC spheroids transmit positive survival signals.

To determine if Netrin dependent signaling is active in HGSOC spheroids, we stimulated cells in suspension with recombinant Netrin-1 (*Figure 5C*). Western blotting for phospho-ERK, as a measure of activation, revealed stimulation of MEK-ERK signaling in each of the cell lines tested. Because MAPK signaling is best known for its role in proliferation, we also tested Netrin-1 stimulation of quiescent, adherent cells and compared this with stimulation in spheroid culture. Both conditions display similar, low levels of phospho-ERK, but Netrin-1 was only able to stimulate ERK phosphorylation in suspension (*Figure 5D*). Importantly, stimulation of adherent cells with an equivalent concentration of EGF induced pERK levels commonly seen in proliferating cells. Collectively, the context and magnitude of pERK stimulation by Netrin-1 is inconsistent with proliferative signaling, suggesting it may contribute to survival. This suggested that Netrin is a candidate pathway to provide low level, but essential survival signals through the MAPK pathway in dormant spheroids. We investigated the dependence of ERK phosphorylation on UNC5H and DDN receptors and discovered that it is lower when receptors are deleted (*Figure 5E*). Furthermore, Netrin-1 stimulation of suspension cultures of OVCAR8 cells, or derivatives deleted for UNC5 family members or DDN receptors, indicated that Netrin receptors are essential to activate ERK in suspension culture (*Figure 5F*). To confirm that Netrin-MEK-ERK signaling contributes to survival in suspension culture, we used two different MEK inhibitors, PD184352 and Trametinib, to treat OVCAR8 suspension cultures and observed that it causes loss of cell viability in suspension culture (*Figure 5G*). Since we routinely use reattachment of spheroids to adherent plastic

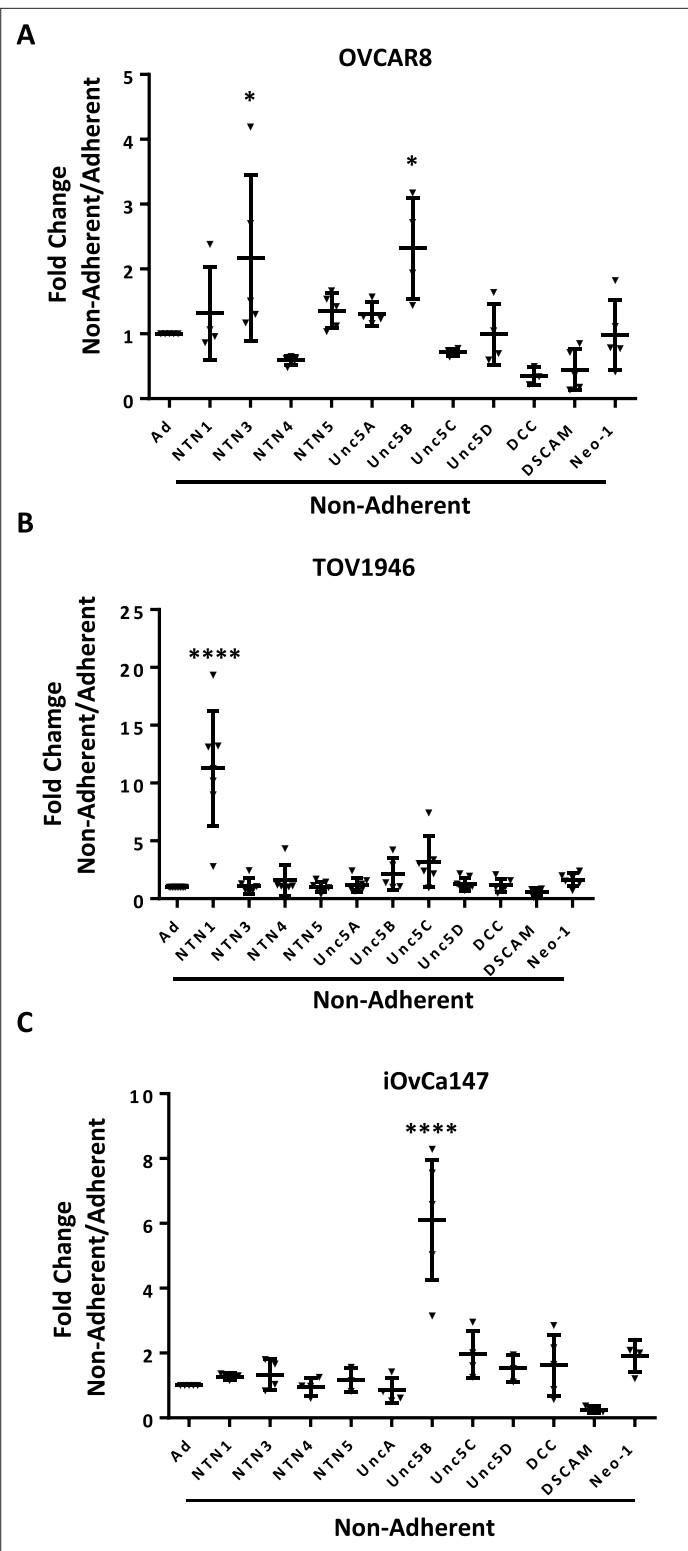

**Figure 3.** Expression of Netrin ligands and their dependence receptors is increased in suspension culture. (**A–C**) RT-qPCR was performed to quantitate mRNA expression levels of Netrin ligands and receptors in three different HGSOC cell lines. Relative expression of the indicated transcripts is shown for suspension culture conditions compared with adherent. All experiments were performed in at least triplicate biological replicates. Means were compared with the same gene in adherent culture using a one way anova (*p<0.05, ****p<0.0001).

The online version of this article includes the following source data for figure 3:

**Source data 1.** Numerical data used in the graph in *Figure 3*.

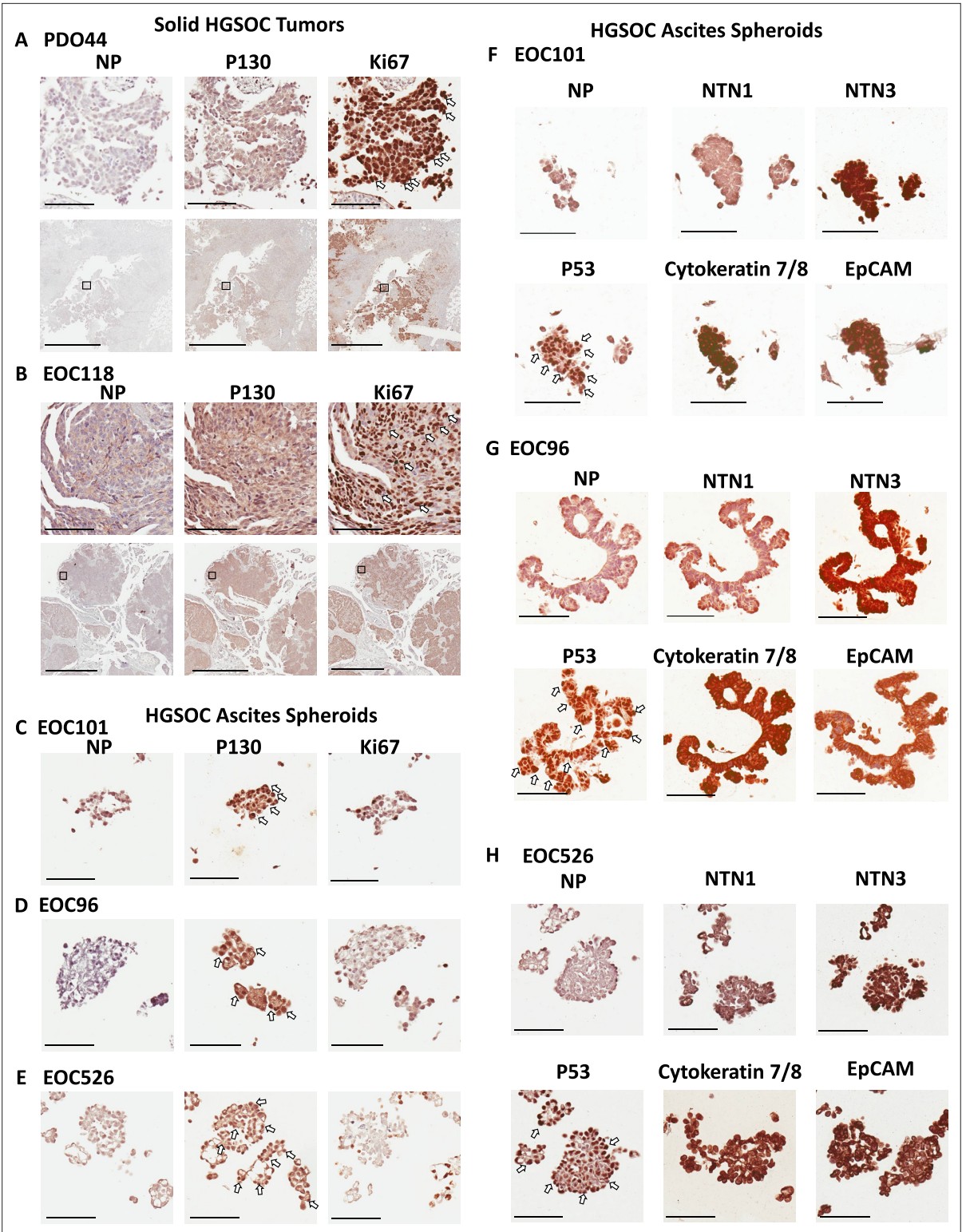

**Figure 4.** Netrins are expressed in dormant patient derived spheroids. Spheroids were isolated from ascites of HGSOC patients and processed for immunohistochemical staining. (**A–B**) Serial sections from solid HGSOC tumors were stained with antibodies to p130, Ki67, or a no primary antibody control (NP). Scale bar = 100 μm for upper panels and 2 mm for lower panels. (**C–E**) Serial spheroid sections from the indicated patients were stained with Ki67 and p130. Sections with dense positive nuclear staining are indicated with white arrows. Scale bar = 100 μm. (**F–H**). Immunohistochemical staining was performed for the indicated proteins on serial sections of ascites derived spheroids. Omission of primary antibody was used a control for background staining for each patient sample (NP). Scale bar = 100 μm.

*Figure 4 continued on next page*

*Figure 4 continued*

The online version of this article includes the following figure supplement(s) for figure 4:

**Figure supplement 1.** Specificity of Netrin-1 and Netrin-3 IHC staining of OVCAR8 spheroids.

as a test of viability, also tested viability using trypan blue dye exclusion to confirm that Trametinib is truly killing cells in suspension and it revealed the same conclusion (*Figure 5H*). As expected, treatment of cells in suspension with Trametinib lead to lower phospho-ERK levels (*Figure 5I*). The simplest explanation of this data is that Netrins signal through a heterodimer or multi-receptor complex containing both an UNC5 and DDN component that provides low level stimulation for MEK and ERK specifically in dormant culture conditions.

## Netrin signaling is essential for dormant spheroid survival during metastatic dissemination

To investigate the role of Netrin signaling in the context of residual disease and relapse, we xenografted mice with control OVCAR8L cells, or UNC5 4KO cells, by intraperitoneal injection (*Figure 6A*). Two weeks following injection, mice were euthanized, and spheroids were obtained by washing the abdominal cavity with PBS. Importantly, at this early stage of engraftment, solid tumor lesions were not yet observable, indicating that we are modeling minimal residual disease where dormancy is most relevant. Extraction of RNA from control OVCAR8 cell spheroids recovered at this time point indicates a reduction in expression of pro-proliferative genes compared to the same cells prior to engraftment (*Figure 6B*). Furthermore, comparison with RT-qPCR measurements of these same markers in suspension culture induced OVCAR8 spheroids indicates that spheroids from abdominal washes are equivalently withdrawn from the cell cycle (*Figure 6*; *Figure 6—figure supplement 1*). This suggests spheroids at this stage are dormant. Furthermore, recovery of spheroids by replating on adherent plastic indicates that deletion of UNC5 receptors to block Netrin signaling extensively reduces survival under these circumstances (*Figure 6C*). We also quantitated cancer cells in these abdominal washes by qPCR using human Alu repeat detection to count cells and this also demonstrates that cancer cell abundance is reduced in this dormant state by loss of UNC5 receptors. Further xenografts were performed to investigate the long term consequences of reduced survival of UNC5 4KO cells in mice. This demonstrated that mice engrafted with UNC5 4KO cells had a considerably longer survival than control OVCAR8 recipient mice that received the same number of cells (*Figure 6E and F*). Collectively, this demonstrates that inhibition of Netrin signaling compromises viability and eventual disease progression in a model of dormancy and dissemination.

## Netrin overexpression is correlated with poor outcomes in HGSOC

Genomic studies indicate that HGSOC is one of the most aneuploid cancers (*Network CGA, 2011*). In addition to the frequency of copy number variants per tumor, the locations of gains and losses are dissimilar between cases, making this a highly heterogenous disease. Evaluation of TCGA genomic data for HGSOC indicates rare amplifications or deletions of Netrin or Netrin-receptor genes (*Figure 7*; *Figure 7—figure supplement 1A*). Because of redundancy of Netrins and their receptor families this data suggests HGSOC cases with deficiency for Netrin signaling are rare. This is consistent with our data that all six cell lines tested reveal evidence of Netrin component dependence for survival (*Figure 5B*). Some TCGA HGSOC cases display elevated gene expression for Netrins or their receptors, but only rare cases display low expression (*Figure 7*; *Figure 7—figure supplement 1B*). This suggests that Netrin signaling in HGSOC is retained despite high copy number variation in this cancer type and there is occasional overexpression.

In agreement with our experimental data that multiple Netrin ligands can contribute to spheroid survival in HGSOC (*Figure 5A and B*), sorting cases by high level expression of Netrin-1 and –3 ligands reveals a significantly shorter overall survival compared to cases with low level expression and this trend is also evident in the cases that only overexpress Netrin-3 (*Figure 7A*). This genomic data supports the interpretation that Netrin signaling is retained in HGSOC and its elevation is deleterious to patient outcome.

We investigated overexpression of Netrin-1 and –3 in spheroid survival by generating OVCAR8 derivatives that stably overexpress these proteins (*Figure 7B*). We then utilized these cells in suspension

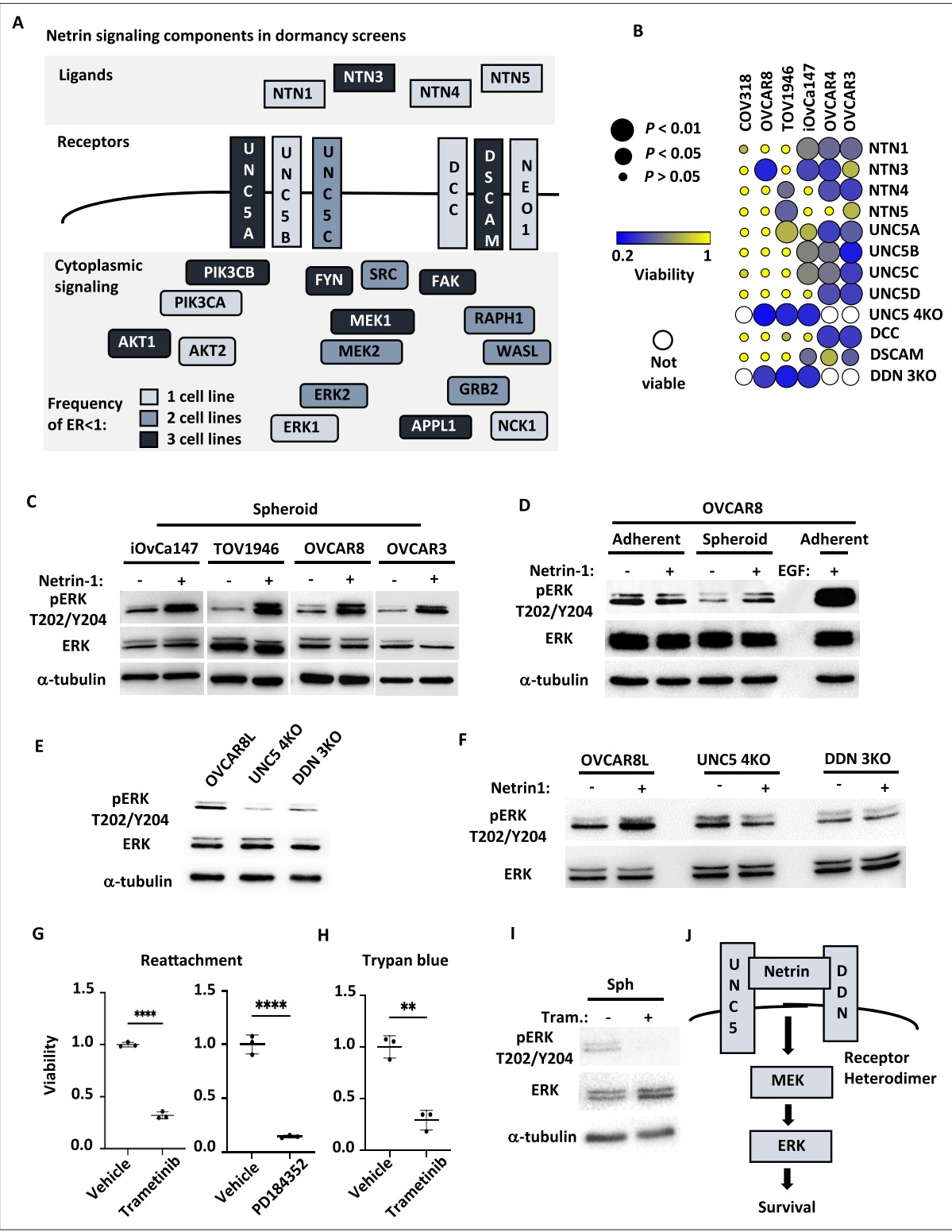

**Figure 5.** Netrin ligands and their receptors are required for spheroid cell survival. (**A**) Illustration of netrin ligands, receptors, and other intracellular signaling molecules that are included in the Axon Guidance pathway category. The frequency of their identification in CRISPR screens is illustrated by shading and indicates how many cell lines had an enrichment ratio <1 for a given component. (**B**) The indicated ovarian cancer cell lines were infected with lentiviruses expressing sgRNAs directed against the indicated Netrin signaling genes. Cells were transferred to suspension culture conditions to induce spheroid formation for 72 hr and then returned to adherent conditions for 24 hr to facilitate reattachment. Re-attached cells were stained with Crystal Violet and retained dye was extracted and quantitated to measure relative survival. Each cell-gene combination was assayed in at least three

*Figure 5 continued on next page*

*Figure 5 continued*

biological replicates, averaged, and viability is displayed as a bubble plot. Mean survival for a given cell-gene combination was compared with GFP control gRNA transduced cells using one way anova and significance levels are illustrated by bubble size. Inviable cell-gene combinations are depicted as empty spaces. (**C**) Cultures of the indicated cell lines were cultured in suspension for five days and stimulated with 0.5 µg/mL Netrin-1. Netrin-1 signaling was analyzed by SDS-PAGE and western blotting for phospho-ERK, ERK, and tubulin. (**D**) Quiescent adherent OVCAR8 cells were stimulated with 0.5 µg/mL Netrin-1 or 0.5 µg/mL EGF, and compared with OVCAR8 cells in suspension stimulated as in C. Extracts were prepared and blotted for phospho-ERK, ERK, and tubulin. (**E**) Suspension cultures of OVCAR8, or knock out derivatives, were harvested and analyzed for relative phosphorylation levels of ERK by western blotting. Total ERK and tubulin blotting serve as expression and loading controls. (**F**) Netrin-1 signaling in OVCAR8, UNC5 4KO and DDN 3KO derivatives was tested by transferring cells to suspension and stimulating with Netrin-1 as before. Western blotting for phospho-ERK, ERK were also as before. (**G**) OVCAR8 cells were seeded in suspension culture and treated with the MEK inhibitors PD184352, Trametinib or DMSO vehicle for 72 hr. Mean viability was determined by re-attachment and compared by one way anova (****p<0.0001). (**H**) OVCAR8 cells were cultured in suspension and treated with Trametinib or DMSO vehicle as in G. Viability was determined by trypan blue dye exclusion and compared by one way anova (**p<0.01). (**I**) Extracts were prepared from Trametinib and control treated spheroid cells and blotted for phospho-ERK, ERK, and tubulin. (**J**) Model summarizing the roles of Netrin ligands, receptors, and downstream targets MEK and ERK in dormant survival signaling.

The online version of this article includes the following source data and figure supplement(s) for figure 5:

**Source data 1.** Numerical data used for bubble plot in *Figure 5B*.

**Source data 2.** Original files for western blot analysis in *Figure 5C* (pERK T202/Y204, ERK, Tubulin).

**Source data 3.** PDFs containing annotation of original western blots in *Figure 5C* (pERK T202/Y204, ERK, Tubulin).

**Source data 4.** Original files for western blot analysis in *Figure 5D* (pERK T202/Y204, ERK, Tubulin).

**Source data 5.** PDFs containing annotation of original western blots in *Figure 5D* (pERK T202/Y204, ERK, Tubulin).

**Source data 6.** Original files for western blot analysis in *Figure 5E* (pERK T202/Y204, ERK, Tubulin).

**Source data 7.** PDFs containing annotation of original western blots in *Figure 5E* (pERK T202/Y204, ERK, Tubulin).

**Source data 8.** Original files for western blot analysis in *Figure 5F* (pERK T202/Y204, ERK).

**Source data 9.** PDFs containing annotation of original western blots in *Figure 5F* (pERK T202/Y204, ERK).

**Source data 10.** Numerical data used for graphs in *Figure 5G and H*.

**Source data 11.** Original files for western blot analysis in *Figure 5E* (pERK T202/Y204, ERK, Tubulin).

**Source data 12.** PDFs containing annotation of original western blots in *Figure 5E* (pERK T202/Y204, ERK, Tubulin).

**Figure supplement 1.** Evaluation of target gene transcript levels in multigene knock out cells.

**Figure supplement 1—source data 1.** Numerical data used for graphs in *Figure 5—figure supplement 1A–C*.

**Figure supplement 1—source data 2.** Original files for western blot analysis in *Figure 5—figure supplement 1D* (pDAPK1 S318, DAPK1, Tubulin).

**Figure supplement 1—source data 3.** PDF containing annotation of original western blots in *Figure 5—figure supplement 1D* (pDAPK1 S318, DAPK1, Tubulin).

culture assays for spheroid survival as before (*Figure 7C*). This data indicates that overexpression of either Netrin-1 or –3 increases OVCAR8 cell survival in dormant spheroid culture conditions. Examination of fixed spheroids isolated from this experiment further indicates that elevated survival in this experiment is explained by an increase in abundance of individual spheroids that overexpress either Netrin-1 or –3 (*Figure 7D*). These experiments reveal the significance of a previously undiscovered role for Netrins in HGSOC. High level expression correlates with poor outcome and in suspension culture, overexpression confers oncogenic properties on HGSOC cells.

## Netrin overexpression induces abdominal spread of HGSOC

Our findings from CRISPR screens and cell culture experiments indicate that Netrin signaling supports dormant HGSOC cell survival. Our analysis of TCGA is also predictive of a worse outcome for patients. We sought to solidify these findings by searching for how Netrin signaling affects HGSOC disease pathogenesis.

We utilized a xenograft assay in which OVCAR8 cells overexpressing Netrin-1 or –3 were injected into the peritoneal cavity and compared with OVCAR8 cells bearing an empty vector as control. Mice were harvested following 35 days to analyze disease dissemination and characterize the effect of Netrins in this model of HGSOC metastatic spread (*Figure 8A*). Necropsy of all animals at endpoint allowed for the identification and quantitation of tumor nodules within the abdomen (*Figure 8B*). Petal plots illustrate the frequency of specific organ and tissue sites displaying tumor nodules (*Figure 8B*). This revealed that spread to the omentum was similar across all genotypes of xenografted cells and it

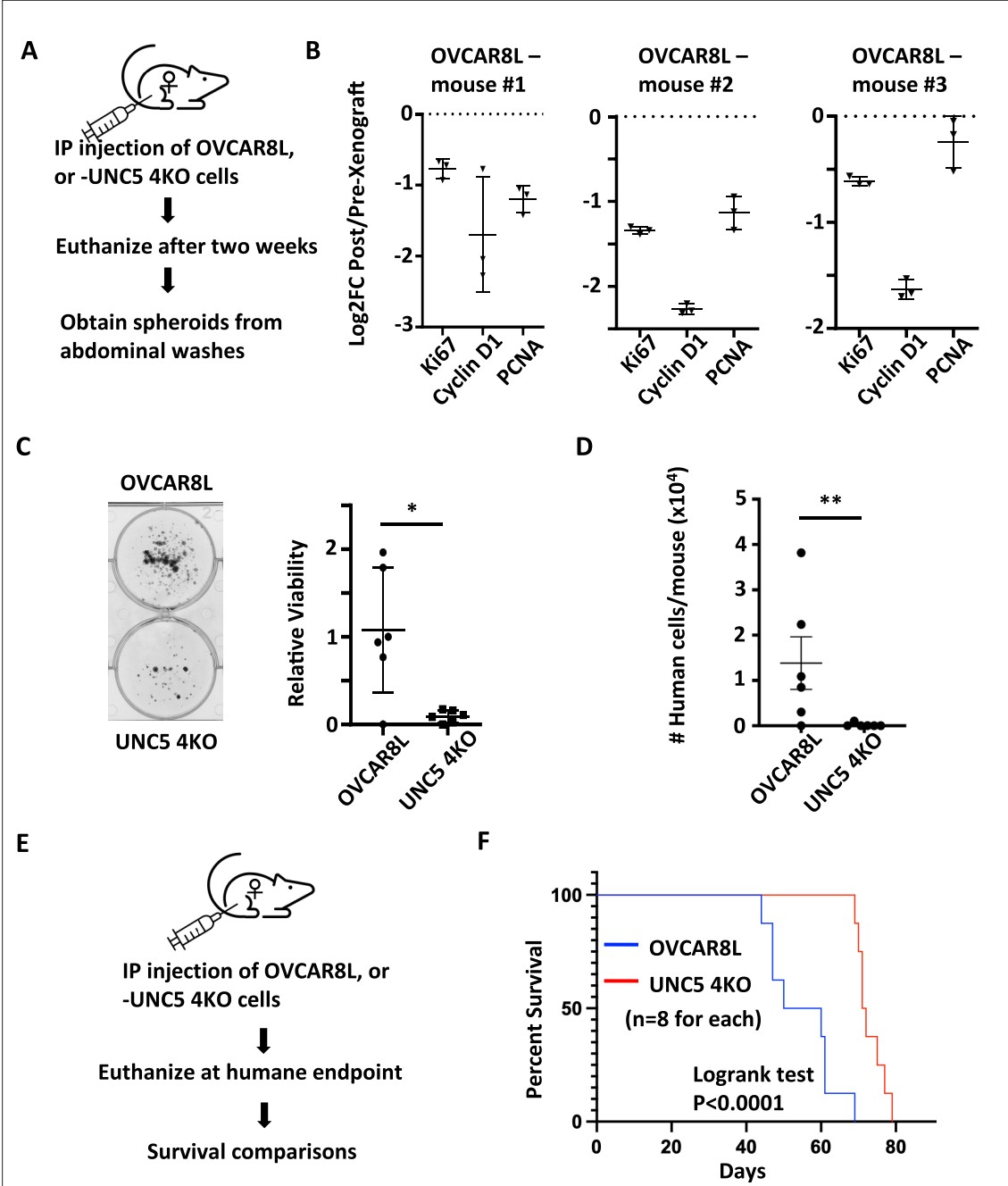

**Figure 6.** Loss of Netrin signaling reduces spheroids and prolongs survival. (**A**) Control OVCAR8L (negative control CRISPR targeted cells) and UNC5 4KO cells were injected into the peritoneal space of female NOD/SCID mice. After 2 weeks, animals were euthanized and engrafted cells were collected with abdominal washes. (**B**) RT-qPCR was used to compare gene expression of the indicated cell cycle markers in RNA extracted from proliferating cells in culture before engraftment and compared with RNA obtained from spheroids in xenografted mice. Technical replicates of RNA derived from three different mice is shown. (**C**) Spheroids obtained from abdominal washes from each mouse were plated in adherent conditions to compare their abundance between control and UNC5 4KO genotypes. One example of each genotype of cell is shown. Crystal violet staining and dye extraction were used to quantitate biomass and averages were compared by one-way anova (n=6, *p<0.05). (**D**) DNA was extracted from cells collected in abdominal washes and human Alu repeats were detected by qPCR to quantitate and compare human cancer cells. Means were compared by one-way anova (n=6, **p<0.01). (**E**) Control OVCAR8 and UNC5 4KO cells were xenografted as above and mice were monitored until humane endpoint. (**F**) Kaplan-Meier analysis of survival for mice engrafted with the indicated genotypes of cells. Survival was compared by logrank test.

The online version of this article includes the following source data and figure supplement(s) for figure 6:

**Source data 1.** Numerical data used for graphs in *Figure 6B*.

*Figure 6 continued on next page*

*Figure 6 continued*

**Source data 2.** Numerical data used for graphs in *Figure 6C*.

**Source data 3.** Numerical data used for graphs in *Figure 6D*.

**Source data 4.** Numerical data used for graphs in *Figure 6F*.

**Figure supplement 1.** Evaluation of dormancy arrest in cell culture by RT-qPCR.

**Figure supplement 1—source data 1.** Numerical data used for graphs in *Figure 6—figure supplement 1*.

occurred in all mice (*Figure 8B*). In contrast, tumor nodules were detected with increased abundance on the diaphragm, the liver, and mesenteries that support the uterus (mesometrium; *Figure 8B*). In addition to more frequent disease spread to these locations caused by Netrin overexpressing cells, we note that the quantity of nodules was also increase in comparison with OVCAR8 control cells (*Figure 8C–F*). We confirmed the source of these lesions to be the xenografted cells using immuno-histochemical staining for human cytokeratin (*Figure 8G and H*). Lastly, Netrin-1 and –3 histochemical staining confirms its presence in the local tumor microenvironment (*Figure 8G and H*). The similarity of increased tumor nodules seen in xenografts and elevated spheroid formation and viability seen in culture suggests that Netrin signaling improves cell viability in spheroid aggregates in vivo leading to increased disease spread.

## Discussion

In this report, we investigated survival mechanisms in a model of HGSOC dormancy. We utilized a modified CRISPR screening approach that was adapted to discover gene loss events in largely non-proliferative conditions. We compared essential genes discovered in the screen with gene expression increases identified by RNA-seq in dormant HGSOC cultures. This implicated the Netrin signaling pathway in HGSOC spheroid survival. We independently validated the role of Netrins, their receptors, and downstream intracellular targets MEK and ERK in the survival biology of HGSOC cells in dormant culture and in vivo in xenografts. Furthermore, we reveal that overexpression of Netrin-1 and –3 are correlated with poor clinical outcome in HGSOC and over expression of these ligands induces elevated survival in dormant HGSOC culture conditions and increased disease spread in a xenograft model of dissemination. Our work implicates Netrins, their receptors, and MEK-ERK as a previously unappreciated, but critical signaling network in HGSOC survival.

HGSOC spheroid cell biology has presented a unique challenge to cancer chemotherapy. Dormant spheroids are most abundant in patients with late-stage disease and they contribute to therapeutic resistance and seed metastases (*Ahmed and Stenvers, 2013*; *Narod, 2016*; *Keyvani et al., 2019*). Hence, there is a critical unmet need to understand HGSOC spheroid dependencies and target them therapeutically. Our data suggests that Netrin signaling is a universally active mechanism in spheroid cell survival. HGSOC is characterized by extensive genomic rearrangements and copy number variants (*Network CGA, 2011*), and we speculate that Netrin signaling is retained because of the extensive redundancy of components of this pathway. Netrin-1 and –3 can both contribute to survival signaling and knock out of three or four different Netrin receptors is necessary to compromise Netrin signaling in survival. A key goal for the future is to identify Netrin-1 or –3 overexpression patients and compare their specific disease progression patterns with our xenograft model. This will be most informative of the effects of Netrin ligand overexpression in scenarios of dormancy, metastasis, and chemotherapy resistance.

This study demonstrates that Netrins function to stimulate UNC5H and fibronectin repeat containing receptors such as DCC to activate survival signals in HGSOC spheroids. This is distinct from the known role of UNC5H and DCC as dependence receptors where Netrin inhibits their pro-apoptotic signaling (*Brisset et al., 2021*). Our data does not show elevated survival of spheroid cells in response to dele-tions of receptors, even in instances where families of Netrin receptors are co-deleted. Consistent with this observation, over expression of Netrin-1 or –3 in OVCAR8 cells stimulates higher level survival in suspension. In colon cancer, deletion of DCC confers a survival advantage and underscores the depen-dence receptor concept (*Fearon et al., 1990*; *Mehlen et al., 1998*). In HGSOC DCC or UNC5H dele-tions are relatively rare (*Network CGA, 2011*). This suggests that ovarian cancer cells utilize Netrin signals for positive survival signaling instead of to neutralize inherent death signals from dependence

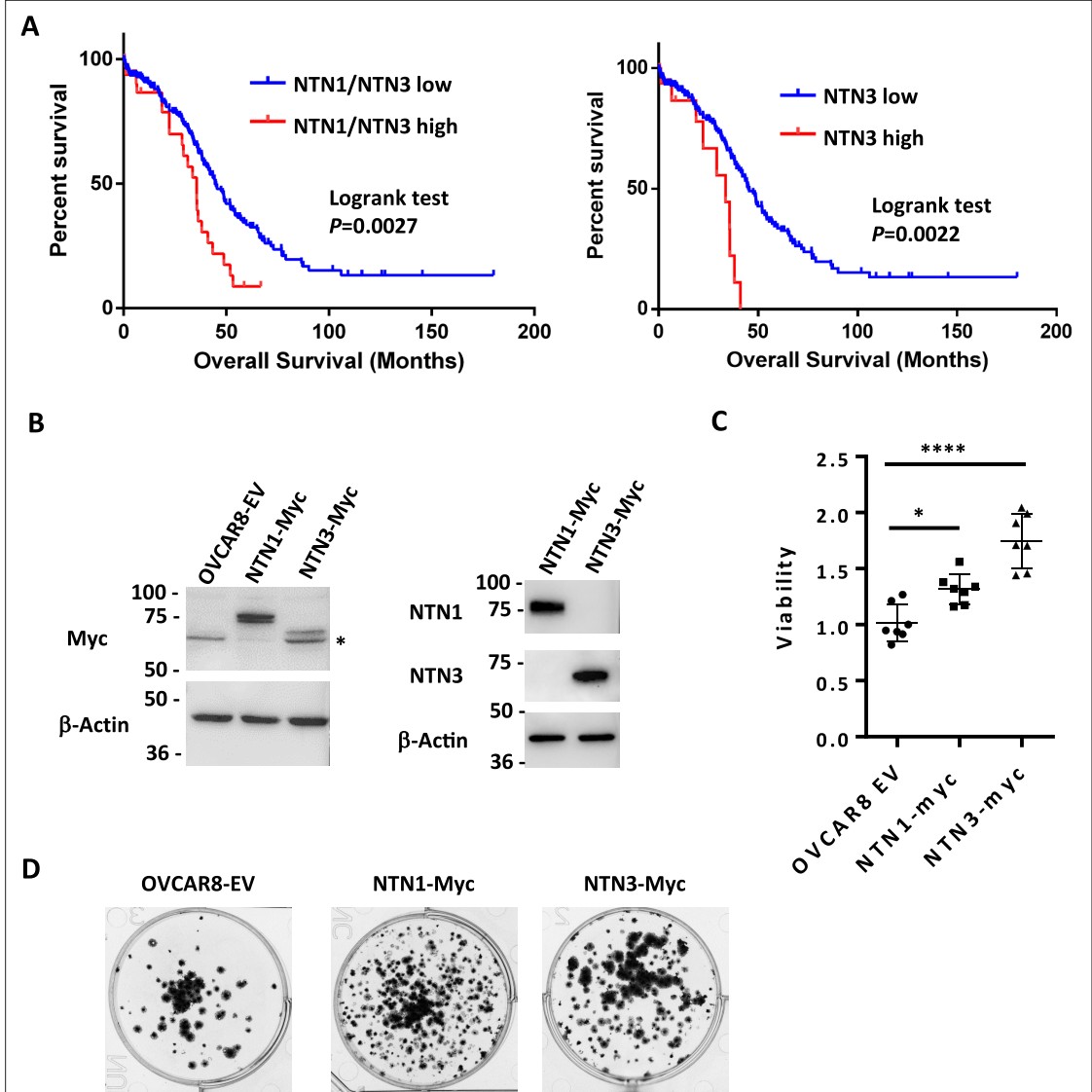

**Figure 7.** Netrin ligand overexpression is associated with poor clinical outcome in HGSOC. (**A**) TCGA RNA-seq data for HGSOC patients (TCGA PanCancer Atlas study) was used to identify high Netrin-1 or –3 and low Netrin-1 or –3 expressing patients (high expressing are above z-score 1.2). Overall survival was used to construct Kaplan-Meier plots and survival was compared using a logrank test. (**B**) OVCAR8 cells were stably transduced with lentiviral constructs to overexpress epitope tagged Netrin-1 or –3. Western blotting for Netrins, Myc-tags, and Actin were used to determine relative expression levels of both Netrins in these cell populations and vector controls. (**C**) Control and Netrin overexpressing cells were transferred to suspension culture conditions to form spheroids, and replated to assay for viability. Mean viability and standard deviation is shown for each. One-way anova was used to compare survival (* p<0.05, *** p<0.001). (**D**) Reattached spheroids were fixed and stained with Crystal Violet to examine size and abundance in control and Netrin-1 or –3 overexpression.

The online version of this article includes the following source data and figure supplement(s) for figure 7:

**Source data 1.** Original files for western blot analysis in *Figure 7B* (Myc, Netrin-1, Netrin-3, Actin).

**Source data 2.** PDFs containing annotation of original western blot analysis in *Figure 7B* (Myc, Netrin-1, Netrin-3, Actin).

**Source data 3.** Numerical data used for graphs in *Figure 7C*.

**Figure supplement 1.** Frequency of Netrin ligand and receptor deletions, mutations or expression changes in high grade serous ovarian cancer.

receptors. Our data suggests that Netrin activates MEK and ERK to support cell survival indicating that targeting ligands, receptors or downstream at MEK are all possible approaches to inhibit this survival pathway. Netrin-1 inhibition in solid tumor xenograft models of melanoma have been shown to be effective in combination with chemotherapy (*Boussouar et al., 2020*), but its effects on a

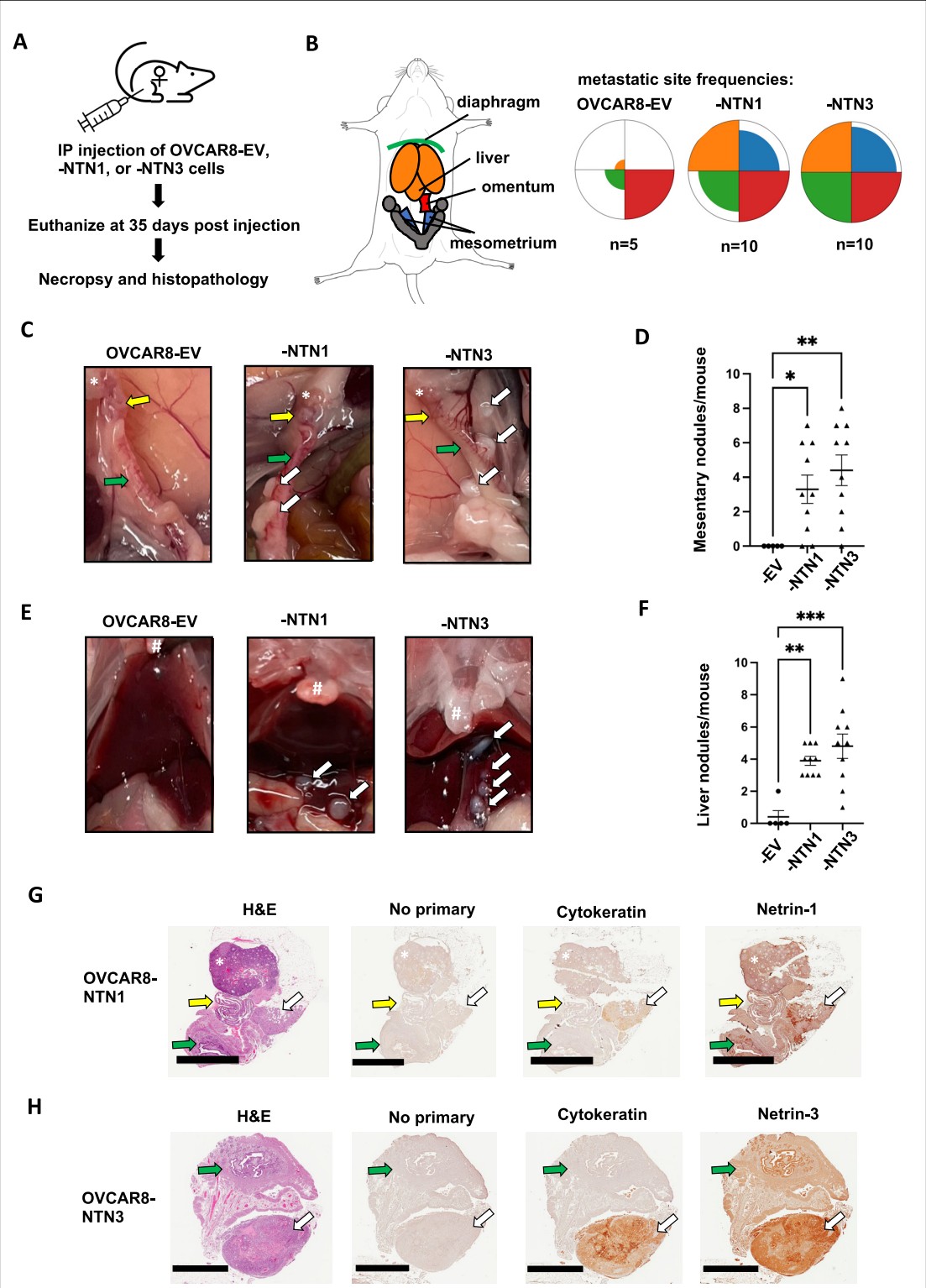

**Figure 8.** Netrin overexpression causes increased dissemination of tumor nodules. (**A**) Netrin overexpressing and control OVCAR8 cells were injected into the intraperitoneal space of female NOD/SCID mice. Mice were euthanized following 35 days and analyzed for disease burden by necropsy and histopathology. (**B**) The spread of cancer to the diaphragm, liver, omentum, and mesometrium was determined from necropsies and colors used in the anatomical schematic correspond with petal plots for each genotype of cells. Petal plot radius illustrates frequency of mice bearing disease spread to a particular location. The radius of color fill is proportional to the total number of animals with tumor nodules found in that location. (**C**) Photographs of necropsy findings in the mesometrium. Locations of ovaries (*), oviduct (yellow arrow), and uterine horn (green arrow) are indicated in each case. Tumor

*Figure 8 continued on next page*

*Figure 8 continued*

nodules are indicated by white arrows. (**D**) The number of mesometrium associated tumor nodules was determined for each mouse. Mean values are indicated and differences between genotype were determined by one way anova (* p<0.05, ** p<0.01). (**E**) Photographs of necropsy findings in the liver. Location of sternum is indicated (#) in each image. Tumor nodules are indicated by white arrows. (**F**) The number of liver associated tumor nodules was determined for each mouse. Mean values are indicated and differences between genotype were determined by one way anova (** p<0.01, *** p<0.001). (**G** and **H**) Histology of mesometrial tumor nodules are shown. Serial sections were stained with H&E, or with the indicated antibodies for immunohistochemistry. Ovaries are indicated (*), as are the oviduct (yellow arrow), the uterus (green arrow), and the tumor nodule (white arrow). Scale bar = 2 mm.

The online version of this article includes the following source data for figure 8:

**Source data 1.** Numerical data used for graphs in *Figure 8*.

dormant scenario are unknown. Identifying agents that cooperate with standard HGSOC chemotherapeutic agents such as carboplatin would offer an attractive strategy to eradicate dormant cells while combating progressive disease. Alternatively, a novel approach to targeting dormancy discovered in this study is, ironically, MEK inhibition. While MAPK signaling in cancer is universally known for its role in proliferation (*Downward, 2003*), and MEK inhibitors are used to block proliferation (*Barbosa et al., 2021*; *Flaherty et al., 2012*), it is noteworthy that Netrin stimulation of MAPK signaling is reported to provide survival signals and to guide axon outgrowth in a non-proliferative context (*Forcet et al., 2002*; *Chen et al., 2017a*). Dormancy is characterized by low phospho-ERK levels (*Sosa et al., 2014*; *Yeh and Ramaswamy, 2015*), indicating low MEK activity, however low levels of RAS-MAPK signaling are known to provide survival signals (*Downward, 2003*). A challenge with deploying MEK inhibitors as an anti-dormancy setting in HGSOC therapy is its incompatibility with standard chemotherapies that rely on cell proliferation. Overall, our study aimed to identify potential new therapeutic targets in dormancy and Netrins and MEK are potential new candidates for dormant HGSOC disease.

Current literature offers little to connect Netrins with HGSOC beyond the observation that *NTN1* expression is increased in tumors compared to benign lesions (*Papanastasiou et al., 2011*). Similarly, dormancy literature does not include a role for Netrins (*Sosa et al., 2014*; *Phan and Croucher, 2020*; *Goddard et al., 2018*; *Yeh and Ramaswamy, 2015*; *Recasens and Munoz, 2019*). However, we expect that Netrin signaling may play an unappreciated role in dormancy in other cancer disease site paradigms. As stated in the introduction, dormancy is characterized by acquisition of stem like properties, EMT, and in most paradigms it involves rare cells establishing themselves in perivascular or bone marrow niches and Netrins have also been implicated in these processes (*Renders et al., 2021*; *Ozmadenci et al., 2015*; *Lengrand et al., 2023*; *Ducarouge et al., 2023*; *Cassier et al., 2023*). Inhibition of Netrin-1 with NP137 and treatment with decitabine has been shown to reduce lung seeding suggesting that Netrins are relevant in a metastatic scenario that may include dormant intermediate steps in dissemination (*Grandin et al., 2016*). Furthermore, Netrin signaling through UNC5B is known to direct vascular branching (*Lu et al., 2004*). This suggests that Netrins are available in the perivascular niche where dormant cells often reside. Lastly, Netrin-1 has been shown to be a component of the bone marrow niche that suppresses hematopoietic stem cell proliferation (*Renders et al., 2021*). While not a cancer dormancy scenario, it indicates that Netrins support some solitary dormant cell properties and are expressed by resident cells in the appropriate dormant cell niches beyond just HGSOC. For these reasons we expect Netrins will have a significant role in tumor dormancy beyond ovarian cancer.

## Methods
### Cell lines and engineering Cas9 +cells

High-grade serous ovarian cancer (HGSOC) cell lines OVCAR3, OVCAR4, OVCAR8, COV318, TOV1946, and iOvCa147 were used in this study. The iOvCa147 cells have previously been characterized and reported (*Tong et al., 2017*). All cell lines were verified by STR analysis. iOvCa147, OVCAR8, and TOV1946 cells were engineered to express Cas9 using pLentiCas9-Blast and clones were isolated. High Cas9 editing efficiency was determined by viability studies using sgRNAs targeting selected fitness genes (PSMD1, PSMD2, EIF3D) and a non-targeting control (LacZ) as previously described (*Hart et al., 2015*).

## Lentiviral sgRNA library preparation

HEK293T cells were transfected with the combined A and B components of the GeCKO v2 whole genome library (123,411 sgRNAs in total) along with plasmids encoding lentiviral packaging proteins (*Sanjana et al., 2014*). Media was collected 2–3 days later, cells and debris were pelleted by centrifugation, and supernatant containing viral particles was filtered through a 0.45 μM filter and stored at −80 °C with 1.1 g/100 mL BSA.

## GO-CRISPR screens in iOvCa147, OVCAR8, and TOV1946 cells

Cas9-positive and Cas9-negative cells were separately transduced with lentiviruses at a multiplicity of infection of 0.3 and with a predicted library coverage of >1000-fold. Cells were selected in puromycin and maintained in complete media (DMEM/F12 with 10% FBS, 1% pen-strep-glutamine), containing puromycin in all following steps. More than $10^9$ cells were collected and split into three groups consisting of approximately $3.0 \times 10^8$ cells. Triplicate samples of $6.2 \times 10^7$ cells were saved for sgRNA sequence quantitation ($T_0$). The remaining cells were seeded at $2.0 \times 10^6$ cells/mL in 20x10 cm ULA plates. Following 2 days of culture, media containing spheroids was transferred to 10x15 cm adherent tissue culture plates. The next day unattached spheroid cells were collected and re-plated onto additional 15 cm plates. This process was repeated for a total of 5 days to maximally capture viable cells. Attached cells were collected and pooled for DNA extraction ($T_f$). Genomic DNA was extracted using the QIAmp DNA Blood Maxi Kit (Qiagen, #51194). PCR amplification and barcoding performed as described (Hart paper, Cell, 2015) using NEBNext Q5 Ultra mastermix (NEB, # MO544). PCR products were extracted from agarose using a Monarch DNA Gel Extraction Kit (NEB, # T1020), quantitated and next generation sequencing was performed using an Illumina NextSeq 75 cycle High Output kit on an Illumina NextSeq platform.

## TRACS analysis

The library reference file containing a list of all sgRNAs and their sequences (CSV file), raw reads for the pooled sgRNA library FASTQ files (L0) and raw reads (FASTQ files) for all $T_0$ and $T_f$ timepoints and replicates for Cas9-positive and Cas9-negative cells were loaded into TRACS (https://github.com/developerpiru/TRACS, copy archived at *developerpiru, 2020*; *Perampalam et al., 2020*). TRACS then automatically trimmed the reads using Cutadapt (v1.15), built a Bowtie2 (v2.3.4.1) index and aligned the reads using Samtools (v1.7). TRACS used the MAGeCK read count function (v0.5.6) to generate read counts and incremented all reads by 1 to prevent zero counts and division-by-zero errors. The TRACS algorithm was then run using this read count file to determine the Library Enrichment Score (ES), Initial ES, Final ES and the Enrichment Ratio (ER) for each gene. VisualizeTRACS was used to generate graphs of Initial and Final ratios.

## Pathway enrichment analyses

For GO-CRISPR screens and multi-omics comparisons, we used the filtered list of genes obtained using TRACS and performed gene ontology and pathway enrichment analysis using the ConsensusPathDB enrichment analysis test (Release 34 (15.01.2019), http://cpdb.molgen.mpg.de/) for top-ranked genes of interest. $p_{adj}$ values and ER values for each gene were used as inputs. The minimum required genes for enrichment was set to 45 and the FDR-corrected $p_{adj}$ value cutoff was set to <0.01. The Reactome pathway dataset was used as the reference. For each identified pathway, ConsensusPathDB provides the number of enriched genes and a q value ($p_{adj}$) for the enrichment. Bar plots were generated in R 3.6.2 using these values to depict the significant pathways identified.

## Generating DYRK1A knockout cells

A double cutting CRISPR/Cas9 approach with a pair of sgRNAs (sgRNA A and B) was used to completely excise exon 2 (322 bp region) of *DYRK1A* using a px458 vector (Addgene #48138) that was modified to express the full CMV promoter. PCR primers that flank the targeted region of *DYRK1A* were used to verify deletion. Single-cell clones were generated by liming dilutions and evaluated for DYRK1A status by PCR, western blot, and sequencing. See Key resource table for sgRNA and PCR primer sequences.

## DYRK1A IP kinase assay

Whole cell lysates from adherent parental iOvCa147 cells and *DYRK1A*⁻/⁻ cells were extracted using complete RIPA buffer and incubated overnight with DYRK1A antibody (Cell Signaling Technology anti-DYRK1A rabbit antibody #8765). Samples were then washed with buffer and Dynabeads (Thermo Fisher Scientific Dynabeads Protein G #10003D) were added for 2 hr. Samples were then washed with buffer and recombinant Tau protein (Sigma recombinant Tau protein #T0576) and ATP (Sigma #A1852) were added and samples were incubated for 30 min at 37 °C. 5 x SDS was then added and samples were resolved by SDS-PAGE. Antibodies used for western blotting were phosphospecific Tau antibody (Cell Signaling Technology phospho-Tau Ser-404 rabbit antibody #20194) and Tau protein (Cell Signaling Technology anti-Tau rabbit antibody #46687).

## RNA preparation and RNA-sequencing

Total RNA was collected from parental iOvCa147 cells and *DYRK1A*⁻/⁻ cells from 24 hr adherent or 6 hr spheroid conditions using the Monarch Total RNA Miniprep Kit (NEB #T2010S) as described above. RNA was collected in three replicates for each condition (adherent and spheroids). Samples were quantitated using Qubit 2.0 Fluorometer (Thermo Fisher Scientific) and quality control analysis using Agilent 2100 Bioanalyzer (Agilent Technologies #G2939BA) and RNA 6000 Nano Kit (Agilent Technologies #5067–1511). Ribosomal RNA removal and library preparation was performed using ScriptSeq Complete Gold Kit (Illumina #BEP1206). High-throughput sequencing was performed on an Illumina NextSeq 500 platform (mid-output, 150-cycle kit).

## RNA-seq analyses

Raw FASTQ data was downloaded from Illumina BaseSpace. Reads were aligned using STAR 2.6.1 a *Dobin et al., 2013* to the human genome (Homo_sapiens.GRCh38.dna.primary_assembly.fa; sjdbGTF-file: Homo_sapiens.GRCh38.92.gtf) to generate read counts. DESeq2 was run with BEAVR (*Perampalam and Dick, 2020*) and used to analyze read counts (settings: False discovery rate (FDR): 10%; drop genes with less than 1 reads) to identify differentially expressed genes ($\log_2$ fold change >1 for upregulated genes; or $\log_2$ fold change < –1 for downregulated genes; $p_{adj}$ <0.05) for the following comparisons: parental iOvCa147 adherent cells vs. parental iOvCa147 spheroid cells; or parental iOvCa147 spheroid cells vs. *DYRK1A*⁻/⁻ spheroid cells. For RNA-seq, we formed pathway analyses in BEAVR and downloaded the pathway enrichment table to construct bar plots and determine overlapping pathways using R 3.6.2.

## Western blots

Adherent cells were washed in 1 x PBS and lysed on the plate by the addition of complete RIPA buffer with protease (Cell Signaling Technology #5871 S) and phosphatase (Cell Signaling Technologies #5870 S) inhibitors and incubated for 5–10 min on ice. Spheres were collected by gentle centrifugation at 300 rpm, washed gently in ice cold PBS, centrifuged again and then resuspended in RIPA buffer as above. Samples were then centrifuged at 14,000 RPM at 4 °C for 10 min. The supernatant liquid for each sample was collected and the pellets re-extracted with RIPA containing the above inhibitor cocktails. Following centrifugation, the supernatants for each treatment were collected and pooled. Protein was determined by Bradford assay. To assess the effect of Netrin 1 on downstream signaling, spheres incubated under non adherent conditions for 5 days were challenged with the addition of 500 ng/ml human recombinant Netrin-1 (R&D Systems #6419-N1-025) for a period of 20 min before collection. Lysates were mixed with 5 x SDS loading dye buffer and resolved using standard SDS-PAGE protocols. Antibodies used for blotting were p42/44 MAPK (Erk1/2) (Cell Signaling Technology rabbit antibody #4695), Phospho-p44/42 MAPK(Erk1/2) T202/Y204 (Cell Signaling Technology; rabbit antibody #4370), p38 MAPK (Cell Signaling Technology; rabbit antibody #9212), p38 MAPK T180/Y182 (Cell Signaling Technology; rabbit antibody #9211), p130(RBL2, Santa Cruz Biotechnology, rabbit antibody sc-317 discontinued), Ki67 (Abcam rabbit antibody, # ab16667), DYRK1A (Cell Signaling Technology anti-DYRK1A rabbit antibody #8765), Netrin-1 (Abcam rabbit antibody #ab126729), Netrin-3 (Abcam rabbit antibody #ab185200), α-tubulin (Cell Signaling Technology anti-αTubulin rabbit antibody #2125) and β-actin (Millipore Sigma rabbit antibody #A2066).

## RT-qPCR analysis of gene expression

Cells from a minimum of three biological replicates were triturated off the plate using Versene (0.5 mM EDTA in PBS), collected, counted and $10^6$ cells plated onto Ultra-Low Attachment cell culture dishes

(Corning) to induce sphere formation. The spheres were collected after 24 hr and RNA extracted using the Monarch Total RNA Miniprep Kit (New England Biolabs #T2010S). RNA was extracted from adherent cells directly from the plate. Extracted RNA was quantitated using a NanoDrop and stored at –80° until used. First strand cDNA was generated from isolated RNA using the iScript cDNA Synthesis Kit (Bio-Rad, #1708890) and RT-qPCR performed on the resulting cDNA (diluted 5-fold) using the iQ SYBR Green Supermix (Bio-Rad, #1708882) and following the manufactures instructions. Reactions were performed in 0.2 ml non-skirted low profile 96 well PCR plates (Thermo Fisher Scientific, #AD-0700) using a CFX Connect Real Time System thermo-cycler (Bio-Rad).

## TCGA patient data analyses

We used the cBioPortal for Cancer Genomics (https://www.cbioportal.org; *Cerami et al., 2012*) to analyze Netrin pathway alterations in Ovarian Serous Cystadenocarcinoma patients (TCGA, PanCancer Atlas) for genomic alterations and mRNA expression. OncoPrints were generated for mRNA expression (mean z-score threshold ±1.5), and mutations and putative copy number alterations for the following genes: *NTN1, NTN3, NTN4, NTN5, UNC5A, UNC5B, UNC5C, UNC5D, DCC, DSCAM*. Survival analysis of *NTN1*, and *–3* utilized patients from TCGA, PanCancer Atlas dataset and RNA-seq expression (high expression z-score threshold >1.2 and low is below 1.2).

## Generation of gene knockout cell lines

Gibson Assembly (NEB #E2611) was used to clone sgRNAs sequences derived from the GeCKO v2 library per gene of interest (*NTN1, NTN3, NTN4, NTN5, DSCAM, DCC, NEO1, UNC5A, UNC5B, UNC5C, UNC5D,* and *EGFP* control (see Key resource table for sgRNA sequences for each)) into lenti-CRISPR v2 (Addgene #52961). For individual gene knockouts pooled libraries of up to four sgRNAs were generated. For multiple gene knockouts pooled libraries of two sgRNAs/gene were constructed. To each 20- nucleotide guide (see Key resource table) was appended the 5' sequence, 5'-TATCTTGT GGAAAGGACGAAACACC-3' and the 3' sequence, 5'-GTTTTAGAGCTAGAAATAGCAAGTTAAAAT-3' (*Wang et al., 2016*). A 100 bp fragment was then generated using the forward primer, 5'-GGCTTTAT ATATCTTGTGGAAAGGACGAAACACCG-3', and reverse primer, 5'-CTAGCCTTATTTTAACTTGCTATT TCTAGCTCTAAAAC-3'. The resulting gel purified fragment was quantitated by Qubit and cloned into the lentiviral vector LentiCRISPR-V2 previously digested with BsmBI (New England Biolabs, # R0739) as described (*Wang et al., 2016*). To generate virus particles, HEK293T cells were transfected with the assembled plasmid along with plasmids encoding lentiviral packaging proteins. Media was collected after 2–3 days and any cells or debris were pelleted by centrifugation at 500 x *g*. Supernatant containing viral particles was filtered through a low protein binding 0.45 μM filter. For each knockout, it was important to ensure that each target cell was transduced with a full complement of guides. To ensure this, the cells were transduced at relatively high MOI. Briefly, 25,000 cells were plated and transduced with fresh HEK293T conditioned media containing virus. While we did not titer the viral preparations for these experiments, previous experience indicated that we could generate sufficiently high virus titers using our protocol. Under these conditions, we found a lack of puromycin sensitive cells following 2–3 days in media containing 2–4 μg/ml puromycin indicating that every cell received at least one virus particle. Multi-gene knock outs were created through multiple rounds of infection with the relevant culture supernatants followed by one round of puromycin selection.

## Generation of gene overexpression cell lines

Plasmids encoding Netrin1 and Netrin3 were obtained from Addgene (Ntn1-Fc-His, #72104 and Ntn3-Fc-His, #72105). Netrin1 was PCR amplified using the forward primer 5'-GCTATCGATATC CCAAACGCCACCATGATGCGCGCTGTGTGGGAGGCGCTG-3' and reverse 5'-GCTATCTCTAGA GGCCTTCTTGCACTTGCCCTTCTTCTCCCG-3' and cloned into the EcoRV/XbaI sites of pcDNA3.1/ myc-HisA. For lentiviral expression, NTN1 was PCR amplified from pcDNA NTN1 using the primers, forward 5'-CACTGTAGATCTCCAAACGCCACCATGATGCGCGCTGTGTGGGAGGCGCTG-3' and, reverse 5'-ACGCGTGAATTCTTATCAACCGGTATGCATATTCAGATCCTCTTCTGAGAT-3'. Netrin 3 was amplified using the forward primer 5'-GCTAGCGCGGCCGCCACCATGCCTGGCTGGCCCTGG-3' and reverse primer 5'-CTGAGATCTAGAAGCAGCACTACAACGACCACGACGTTCACG-3' and cloned into the NotI/XbaI sites of pcDNA3.1/myc-HisA. For lentiviral expression, NTN3 was amplified from pcDNA NTN3 using the forward primer 5'-CTCGAGAGATCTGCGGCCGCCACCATGCCTGGCTGG

CCCTGG-3' and the same reverse primer as for NTN1. In this way both constructs carry a C-terminal Myc tag. Both cDNA's were inserted into the BamHI/EcoRI sites of the lentiviral expression vector FUtdTW (Addgene #22478). Construction of lentiviral particle and transduction into the target cells was as described above.

## Spheroid formation and reattachment viability assays

For each targeted gene or cDNA expressed in iOvCa147, OVCAR3, OVCAR4, OVCAR8, COV318, or TOV1946 cells, spheroid viability was assayed as follows: Three to ten biological replicate cultures of knockout cells were cultured for 72 hours in suspension conditions using ULA plasticware ($2\times10^6$ cells per well) to allow spheroid formation. Addition of therapeutics (Tramatinib, Selleckchem, #S4484) was made at the time of plating. Spheroids were collected and transferred directly to standard plasticware to facilitate reattachment for 24 hr, or dissociated and mixed with trypan blue. Reattached cells were fixed in 25% methanol in PBS for 3 min. Fixed cells were incubated for 30 min with shaking in 0.5% crystal violet, 25% methanol in PBS. Plates were carefully immersed in water to remove residual crystal violet and destained in 10% acetic acid in 1 x PBS for 1 hr with shaking to extract crystal violet from cells. Absorbance of crystal violet was measured at 590 nm using a microplate reader (Perkin Elmer Wallac 1420).

## Xenografts

ARRIVE principles were followed in carrying out xenograft experiments. Female NOD-SCID mice were purchased from JAX (#001303). They were randomly selected prior to being engrafted between six and eight weeks of age with $4\times10^6$ OVCAR8 control or modified derivatives. Five to 10 mice were housed for 14 days, 35 days, or until humane endpoint for each OVCAR8 or modified derivative. Following euthanasia all mice were subjected to a necropsy to analyze disease spread. Tumor lesions were photographed and quantitated by individuals unaware of the genotypes. Abdominal organs and associated tumor lesions were fixed in 10% formalin overnight and transferred to 70% ethanol and later embedded, sectioned and stained as described below.

## Patient derived spheroids

Patient derived spheroids were isolated from ascites fluid obtained from debulking surgical procedures. These HGSOC samples were obtained with permission and institutional research ethics oversight (Project ID #115904). Spheroids were prepared by filtering the ascites fluid through a 100 µm mesh filter and washing the captured spheroids from the filter with PBS. The spheres were collected by gravity on ice in PBS, washed twice with ice cold PBS and allowed to settle on ice. Spheres were then fixed for 20 min in 10% formalin, washed, and resuspended in PBS before being embedded in paraffin.

## Histology and immunohistochemistry

Fixed organs from xenografts or spheroids were embedded in paraffin, sectioned, and stained with H&E using standard methods. For immunohistochemistry, sections were deparaffinized by standard procedures and antigen retrieval was performed in 10 mM citrate, pH6.0. The sections were blocked with 10% goat serum/3% BSA/0.3% Triton X-100 and stained with antibodies to Netrin-1 (Abcam #ab126729, 1:200), Cytokeratin (human specific cytokeratin 7 and 8, ZETA Corporation #Z2018ML, 1:200), EpCAM (Cell Signaling Technology, #14452, 1:250), p53 (Cell Signaling Technologies, #2527, 1:125), Netrin-3 (1:250), Ki67 (Cell Signaling Technology, #12202, 1:400) and p130 (RBL2, Santa Cruz Biotechnology, SC-317, 1:150). Netrin-3 antibodies were generated using immunized rabbits and their generation and purification will be published elsewhere. Primary antibodies were followed with a biotinylated anti-rabbit (Vector Labs, # BA-1000) or anti-mouse (Jackson ImmunoResearch, #115-067-003) IgG as a secondary antibody (1:500). Proteins were visualized using DAB chemistry (ImmPACT DAB, Vector Labs, #SK-4105).

## Statistical analyses

Specific statistical tests used are indicated in the figure legends for each experiment. Analysis was done using GraphPad Prism 6.

## Materials availability

All previously unpublished reagents created in this study are available on request from the corresponding author.

# Additional information

### Competing interests

Patrick Mehlen: Founder of Netris Pharma. The other authors declare that no competing interests exist.

### Funding

| Funder | Grant reference number | Author |
|---|---|---|
| Cancer Research Society | 23150 | Frederick A Dick |
| Cancer Research Society | 25021 | Frederick A Dick |
| Ontario Institute for Cancer Research | P.CTIP.966 | Frederick A Dick |
| Canadian Institutes of Health Research | PJT173391 | Frederick A Dick |

The funders had no role in study design, data collection and interpretation, or the decision to submit the work for publication.

### Author contributions

Pirunthan Perampalam, Conceptualization, Formal analysis, Investigation, Methodology, Writing – original draft, Writing – review and editing; James I MacDonald, Conceptualization, Formal analysis, Investigation, Visualization, Methodology, Writing – review and editing; Komila Zakirova, Formal analysis, Visualization, Methodology; Daniel T Passos, Formal analysis, Investigation, Methodology; Sumaiyah Wasif, Formal analysis, Investigation, Visualization; Yudith Ramos-Valdes, Resources, Formal analysis, Investigation; Maeva Hervieu, Patrick Mehlen, Rob Rottapel, Rohann JM Correa, Resources, Methodology; Benjamin Gibert, Conceptualization, Resources, Methodology; Trevor G Shepherd, Conceptualization, Resources, Visualization, Methodology; Frederick A Dick, Conceptualization, Resources, Supervision, Funding acquisition, Investigation, Visualization, Methodology, Writing – original draft, Project administration, Writing – review and editing

### Author ORCIDs

Daniel T Passos ⓘ http://orcid.org/0000-0002-3530-2821
Benjamin Gibert ⓘ http://orcid.org/0000-0002-5295-3124
Frederick A Dick ⓘ https://orcid.org/0000-0002-0047-9985

### Ethics

This work was carried out in accordance with standards set by the Canadian Council on Animal Care. Animal experiments were overseen by the Western University Animal Use Committee protocol 2020-039.

Joint public review: https://doi.org/10.7554/eLife.91766.3.sa1
Author response https://doi.org/10.7554/eLife.91766.3.sa2

# Additional files

### Supplementary files

• MDAR checklist

## Data availability

All sequencing data have been deposited in GEO under accession codes GSE190294 and GSE150246. Source data for graphs and original blots are available in source data files, or from public databases as described in the methods section.

The following datasets were generated:

| Author(s) | Year | Dataset title | Dataset URL | Database and Identifier |
|---|---|---|---|---|
| Perampalam P, McDonald JI, Passos DT, Dick FA | 2023 | Netrin and its dependence receptors are mediators of high-grade serous ovarian cancer cell survival | https://www.ncbi.nlm.nih.gov/geo/query/acc.cgi?acc=GSE190294 | NCBI Gene Expression Omnibus, GSE190294 |
| Perampalam P, McDonald JI, Dick FA | 2023 | GO-CRISPR: a highly controlled workflow to improve discovery of gene essentiality | http://www.ncbi.nlm.nih.gov/geo/query/acc.cgi?acc=GSE150246 | NCBI Gene Expression Omnibus, GSE150246 |

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

# Appendix 1

## Appendix 1—key resources table

| Reagent type (species) or resource | Designation | Source or reference | Identifiers | Additional information |
|---|---|---|---|---|
| Gene (*Homo sapiens*) | NTN1 | GenBank | NCBI: 9423 Gene Cards: GC17P100538 | |
| Gene (*Homo sapiens*) | NTN3 | GenBank | NCBI: 4917 Gene Cards: GC17P100538 | |
| Gene (*Homo sapiens*) | NTN4 | GenBank | NCBI: 59277 Gene Cards: GC12M095657 | |
| Gene (*Homo sapiens*) | NTN5 | GenBank | NCBI: 126147 Gene Cards: GC19M048661 | |
| Gene (*Homo sapiens*) | Unc5A | GenBank | NCBI: 90249 Gene Cards: GC05P185698 | |
| Gene (*Homo sapiens*) | Unc5B | GenBank | NCBI: 219699 Gene Cards: GC10P071212 | |
| Gene (*Homo sapiens*) | Unc5C | GenBank | NCBI: 8633 Gene Cards: GC04M095162 | |
| Gene (*Homo sapiens*) | Unc5D | GenBank | NCBI: 137970 Gene Cards: GC08P035235 | |
| Gene (*Homo sapiens*) | Dyrk1A | GenBank | NCBI: 1859 Gene Cards: GC21P037365 | |
| Gene (*Homo sapiens*) | DCC | GenBank | NCBI: 1630 Gene Cards: GC18P052340 | |
| Gene (*Homo sapiens*) | Neo1 | GenBank | NCBI: 4756 Gene Cards: GC15P073051 | |
| Gene (*Homo sapiens*) | DSCAM | GenBank | NCBI: 1826 Gene Cards :GC21M040010 | |
| Strain, strain background (*Escherichia coli*) | Endura competent cells | Biosearch Technologies | 60242-2 | Electrocompetent cells |
| Strain, strain background (*Escherichia coli*) | NEB10 beta | New England Biolabs | C3019H | High efficiency chemically Competent cells |
| Cell line (*Homo-sapiens*) | OVCAR8 Ovarian cancer cell line | This paper | RRID:CVCL 1629 | Ovarian cancer cell line maintained in *T.Shepherd* lab |
| Cell line (*Homo-sapiens*) | OVCAR3 Ovarian cancer cell line | ATCC | HTB-161 | |
| Cell line (*Homo-sapiens*) | TOV1946 ovarian cancer cell line | This paper | RRID:CVCL 4062 | Ovarian cancer cell line maintained in *R. Rottapel* lab |
| Cell line (*Homo-sapiens*) | iOvCa147 Ovarian cancer cells | This paper | | Primary ovarian cancer cell line maintained in *T. Shepherd* lab |
| Cell line (*Homo-sapiens*) | OVCAR4 Ovarian cancer cell line | Millipore-Sigma | SCC258 | |
| Cell line (*Homo-sapiens*) | COV318 Ovarian cancer cell line | Millipore-Sigma | 07071903-1VL | Primary ovarian cancer cell line |
| Cell line (*Homo sapiens*) | Hek293T | ATCC | CRL-3216 RRID:CVCL_0063 | Human embryonic kidney cells |

*Appendix 1 Continued on next page*

*Appendix 1 Continued*

| Reagent type (species) or resource | Designation | Source or reference | Identifiers | Additional information |
|---|---|---|---|---|
| Biological sample (*Homo sapiens*) | EOC96Ovarian cancer cells | London Health Sciences Centre | | Fresh isolate from patient ascites |
| Biological sample (*Homo sapiens*) | EOC101Ovarian cancer cells | London Health Sciences Centre | | Fresh isolate from patient ascites |
| Biological sample (*Homo sapiens*) | EOC526Ovarian cancer cells | London Health Sciences Centre | | Fresh isolate from patient ascites |
| Antibody | Phosphor-p44/42 MAPK (Erk)Rabbit Monoclonalantibody | Cell Signaling technology | #4370 RRID:AB_2315112 | WB 1:1000 |
| Antibody | p44/42 MAPK(Erk) Rabbit Monoclonalantibody | Cell Signaling technology | #4695 RRID:AB_390779 | WB 1:1000 |
| Antibody | Phosphor-p38MAPKRabbit Monoclonalantibody | Cell Signaling technology | #4511 RRID:AB_2139682 | WB 1:1000 |
| Antibody | p38 MAPKRabbit Monoclonalantibody | Cell Signaling technology | #9215 RRID:AB_331762 | WB 1:1000 |
| Antibody | a-TubulinRabbit Monoclonalantibody | Cell Signaling Technology | #2125 RRID:AB_2619646 | WB 1:1000 |
| Antibody | NTN1Rabbit Monoclonalantibody | Abcam | #Ab126729 RRID:AB_11131145 | 1:1000 for Western blots1:150 for IHC |
| Antibody | p130(RBL2) Rabbit polyclonal | Santa Cruz | Discontinued RRID:AB_632093 | 1:1000 for Western Blots1:150 for IHC |
| Antibody | Ki67Rabbit Monoclonalantibody | Cell Signaling Technology | #12202 RRID:AB_2620142 | 1:500 for IHC |
| Antibody | Ki67Rabbit Monoclonalantibody | Abcam | #ab16667 RRID:AB_302459 | 1:1000 for Western blot |
| Antibody | p53Rabbit Monoclonalantibody | Cell Signaling Technology | #2527 RRID:AB_10695803 | 1:120 for IHC |
| Antibody | EpCAMRabbit Monoclonalantibody | Cell Signaling Technology | #93790 RRID:AB_2800214 | 1:150 for IHC |
| Antibody | Myc (9E10)Mouse Monoclonalantibody | Santa Cruz | #sc-40 RRID:AB_627268 | WB 1:1000 |
| Antibody | NTN3 | Gift from P. Mehlan lab | Not commercially available | 1:1000 for Western Blot1:500 for IHC |
| Antibody | Cytokeratin 7/8Mouse Monoclonalantibody | Zeta Corporation | RRID:AB_11162687 Discontinued | 1:250 for IHC |
| Antibody | Phosphor Tau(S404)Rabbit Monoclonalantibody | Cell Siganling Technology | #20194 RRID:AB_2798837 | WB 1:1000 |
| Antibody | TauRabbit Monoclonalantibody | Cell Signaling Technology | #46687 RRID:AB_2783844 | WB 1:1000 |
| Antibody | DYRK1ARabbit Polyclonalantibody | Cell Signaling Technology | #2771 RRID:AB_915851 | WB 1:1000 |
| Antibody | b-ActinRabbit Polyclonalantibody | Millipore Sigma | #A2066 RRID:AB_476693 | WB 1:1000 |
| Antibody | Goat anti-mouse IgG Biotinylated | Jackson ImmunoResearch | #115-067-003 RRID:AB_2338586 | IHC 1:500 |
| Antibody | Goat anti-rabbit IgG Biotinylated | Vector Laboratories | #BP-9100-50 | IHC 1:500 |
| Recombinant DNA reagent | Human GeCKO Lentiviral sgRNA library V2LentiGuide Puro | Addgene | #1000000049 | Human whole genome CRISPR library |
| Recombinant DNA reagent | Lenti-CAS9-Blast | Addgene | #52962 RRID:Addgene_52962 | Lentiviral vector for delivery of CAS9 |
| Recombinant DNA reagent | pLentiCRISPR-Puro-V2 | Addgene | #52961 RRID:Addgene_52961 | Lentiviral vector for delivery of sgRNA |
| Recombinant DNA reagent | pSPCAS9-(BB)2A-GFP(pX458) | Addgene | #48138 RRID:Addgene_48138 | Vector for delivery of sgRNA |

*Appendix 1 Continued*

| Reagent type (species) or resource | Designation | Source or reference | Identifiers | Additional information |
|---|---|---|---|---|
| Recombinant DNA reagent | NTN1-FC-His | Addgene | #72104 RRID:Addgene_72104 | Plasmid carrying full length Murine NTN1 fused to the FC portion of IgG |
| Recombinant DNA reagent | NTN3-FC-His | Addgene | #72105 RRID:Addgene_72105 | Plasmid carrying full length Murine NTN3 fused to the FC portion of IgG |
| Recombinant DNA reagent | FutdTW | Addgene | #22478 RRID:Addgene_22478 | Lentiviral expression vector |
| Recombinant DNA reagent | FutdTW NTN1 | This paper | | Lentiviral vector expressing full length Murine NTN1 |
| Recombinant DNA reagent | FutdTW NTN3 | This paper | | Lentiviral vector expressing full length Murine NTN3 |
| Recombinant DNA reagent | pcDNA3.1myc/His A | ThermoFisher | V80020 | Mammalian expression vector |
| Recombinant DNA reagent | pcDNA3.1myc/His ANTN1 | This paper | | Mammalian expression vectorfor the expression of NTN1 under G418 selection |
| Recombinant DNA reagent | pcDNA3.1myc/His ANTN3 | This paper | | Mammalian expression vector for the expression of NTN3 under G418 selection |
| Recombinant DNA reagent | LentiCRISPRV2NTN1_A | This paper | | Lentiviral vector for the delivery of, sgRNA targeting human NTN1 |
| Recombinant DNA reagent | LentiCRISPRV2NTN1_B | This paper | | Lentiviral vector for the delivery of sgRNA targeting human NTN1 |
| Recombinant DNA reagent | LentiCRISPRV2NTN1_C | This paper | | Lentiviral vector for the delivery of sgRNA targeting human NTN1 |
| Recombinant DNA reagent | LentiCRISPRV2NTN3_1 | This paper | | Lentiviral vector for the delivery of sgRNA targeting human NTN3 |
| Recombinant DNA reagent | LentiCRISPRV2NTN3_2 | This paper | | Lentiviral vector for the delivery of sgRNA targeting human NTN3 |
| Recombinant DNA reagent | LentiCRISPRV2NTN3_3 | This paper | | Lentiviral vector for the delivery of sgRNA targeting human NTN3 |
| Recombinant DNA reagent | LentiCRISPRV2NTN4_A | This paper | | Lentiviral vector for the delivery of sgRNA targeting human NTN4 |
| recombinant DNA reagent | LentiCRISPRV2NTN4_B | This paper | | Lentiviral vector for the delivery of sgRNA targeting human NTN4 |
| Recombinant DNA reagent | LentiCRISPRV2NTN5_A | This paper | | Lentiviral vector for the delivery of sgRNA targeting human NTN5 |
| Recombinant DNA reagent | LentiCRISPRV2NTN5_B | This paper | | Lentiviral vector for the delivery of sgRNA targeting human NTN5 |
| Recombinant DNA reagent | LentiCRISPRV2NTN5_C | This paper | | Lentiviral vector for the delivery of sgRNA targeting human NTN5 |

*Appendix 1 Continued on next page*

*Appendix 1 Continued*

| Reagent type (species) or resource | Designation | Source or reference | Identifiers | Additional information |
|---|---|---|---|---|
| Recombinant DNA reagent | LentiCRISPRV2Unc5A_1 | This paper | | Lentiviral vector for the delivery of sgRNA targeting human Unc5A |
| Recombinant DNA reagent | LentiCRISPRV2Unc5A_2 | This paper | | Lentiviral vector for the delivery of sgRNA targeting human Unc5A |
| Recombinant DNA reagent | LentiCRISPRV2Unc5A_3 | This paper | | Lentiviral vector for the delivery of sgRNA targeting human Unc5A |
| Recombinant DNA reagent | LentiCRISPRV2Unc5B_1 | This paper | | Lentiviral vector for the delivery of sgRNA targeting human Unc5B |
| Recombinant DNA reagent | LentiCRISPRV2Unc5B_2 | This paper | | Lentiviral vector for the delivery of sgRNA targeting human Unc5B |
| Recombinant DNA reagent | LentiCRISPRV2Unc5B_3 | This paper | | Lentiviral vector for the delivery of sgRNA targeting human Unc5B |
| Recombinant DNA reagent | LentiCRISPRV2Unc5C_1 | This paper | | Lentiviral vector for the delivery of sgRNA targeting human Unc5C |
| Recombinant DNA reagent | LentiCRISPRV2Unc5C_2 | This paper | | Lentiviral vector for the delivery of sgRNA targeting human Unc5C |
| Recombinant DNA reagent | LentiCRISPRV2Unc5C_3 | This paper | | Lentiviral vector for the delivery of sgRNA targeting human Unc5C |
| Recombinant DNA reagent | LentiCRISPRV2Unc5D_1 | This paper | | Lentiviral vector for the delivery of sgRNA targeting human Unc5D |
| Recombinant DNA reagent | LentiCRISPRV2Unc5D_2 | This paper | | Lentiviral vector for the delivery of sgRNA targeting human Unc5D |
| Recombinant DNA reagent | LentiCRISPRV2Unc5D_3 | This paper | | Lentiviral vector for the delivery of sgRNA targeting human Unc5D |
| Recombinant DNA reagent | LentiCRISPRV2DCC_1 | This paper | | Lentiviral vector for the delivery of sgRNA targeting human DCC |
| Recombinant DNA reagent | LentiCRISPRV2DCC_2 | This paper | | Lentiviral vector for the delivery of sgRNA targeting human DCC |
| Recombinant DNA reagent | LentiCRISPRV2DCC_3 | This paper | | Lentiviral vector for the delivery of sgRNA targeting human DCC |
| Recombinant DNA reagent | LentiCRISPRV2Neo1_1 | This paper | | Lentiviral vector for the delivery of sgRNA targeting human Neo1 |
| Recombinant DNA reagent | LentiCRISPRV2Neo1_2 | This paper | | Lentiviral vector for the delivery of sgRNA targeting human Neo1 |
| Recombinant DNA reagent | LentiCRISPRV2Neo1_3 | This paper | | Lentiviral vector for the delivery of sgRNA targeting human Neo1 |
| Recombinant DNA reagent | LentiCRISPRV2DSCAM_1 | This paper | | Lentiviral vector for the delivery of sgRNA targeting human DSCAM |
| Recombinant DNA reagent | LentiCRISPRV2DSCAM_2 | This paper | | Lentiviral vector for the delivery of sgRNA targeting human DSCAM |
| Recombinant DNA reagent | LentiCRISPRV2DSCAM_3 | This paper | | Lentiviral vector for the delivery of sgRNA targeting human DSCAM |

*Appendix 1 Continued on next page*

*Appendix 1 Continued*

| Reagent type (species) or resource | Designation | Source or reference | Identifiers | Additional information |
|---|---|---|---|---|
| Sequence-based reagent | NTN1-AsgRNA | GeCKO genomic CRISPRLibrary V2 | CRISPR guide | 5'-GCAGTCGTCGG CGGCGCTAC-3' |
| Sequence-based reagent | NTN3-AsgRNA | GeCKO genomic CRISPRLibrary V2 | CRISPR guide | 5'-CGACTGTCCGG CCGCCGCAG-3' |
| Sequence-based reagent | NTN4-AsgRNA | GeCKO genomic CRISPRLibrary V2 | CRISPR guide | 5'-CACATTAACGT CGAAGTGAC-3' |
| Sequence-based reagent | NTN5-AsgRNA | GeCKO genomic CRISPRLibrary V2 | CRISPR guide | 5'-ATCGTAGCATG GGTCCGCAG-3' |
| Sequence-based reagent | Unc5A-1sgRNA | GeCKO genomic CRISPRLibrary V2 | CRISPR guide | 5'-CTGTGCTGCG CTCGATCACG-3' |
| Sequence-based reagent | Unc5B-1sgRNA | GeCKO genomic CRISPRLibrary V2 | CRISPR guide | 5'-CGTACAGGCG ATGCGGACGT-3' |
| Sequence-based reagent | Unc5C-1sgRNA | GeCKO genomic CRISPRLibrary V2 | CRISPR guide | 5'-TCCCTTCAGG TGGTCGACAC-3' |
| Sequence-based reagent | Unc5D-1sgRNA | GeCKO genomic CRISPRLibrary V2 | CRISPR guide | 5'-CTTACAGGCTA TGCGCACAG-3' |
| Sequence-based reagent | DCC-1sgRNA | GeCKO genomic CRISPRLibrary V2 | CRISPR guide | 5'-GACTTCCTCG CCTCGTAACC-3' |
| Sequence-based reagent | Neo1-1sgRNA | GeCKO genomic CRISPRLibrary V2 | CRISPR guide | 5'-CGGGCTTTAT CGCTGCGTAG-3' |
| Sequence-based reagent | DSCAM-1sgRNA | GeCKO genomic CRISPRLibrary V2 | CRISPR guide | 5'-ATCGTAGATC TCCTCGCCCG-3' |
| Peptide, recombinant protein | Recombinant Human Netrin-1 | R and D Systems | #6419-N1 | 500 ng/ml |
| Commercial assay or kit | Monarch DNA Gel Extraction Kit | New England Biolabs | #T1020 | |
| Commercial assay or kit | Monarch total RNA Miniprep Kit | New England Biolabs | #T2010 | |
| Commercial assay or kit | Monarch PCR DNA Cleanup kit | New England Biolabs | #T1030 | |
| Commercial assay or kit | NEBNext Ultra II Q5 Master Mix | New England Biolabs | #MO544 | |
| Commercial assay or kit | Monarch Plasmid MiniPrep Kit | New England Biolabs | #T1010 | |
| Commercial assay or kit | DNeasy Blood and Tissue DNA isolation kit | Qiagen | #69504 | |
| Commercial assay or kit | NextSeq 75 Cycle NG sequencing kit | Illumina | #20024906 | |
| Commercial assay or kit | iScript cDNA Synthesis Kit | Bio-Rad | #1708890 | |
| Commercial assay or kit | iQ SYBR Green Supermix | Bio-Rad | #1708882 | |
| Commercial assay or kit | RNA 6000 Nano Kit | Agilent Technologies | #5067-1511 | |
| Commercial assay or kit | ScriptSeq Complete Gold Kit | Illumina | #BEP1206 | |
| Commercial assay or kit | NextSeq 500Mid-Output150 cycles | Illumina | #20024907 | |

*Appendix 1 Continued on next page*

*Appendix 1 Continued*

| Reagent type (species) or resource | Designation | Source or reference | Identifiers | Additional information |
|---|---|---|---|---|
| Chemical compound, drug | Trematinib | Selleckchem | #S4484 | |
| Chemical compound, drug | Polybrene | Santa Cruz | #sc-134220 | 8 mg/ml |
| Chemical compound, drug | Crystal Violet | Millipore Sigma | #C6158 | 0.50% |
| Chemical compound, drug | Protease inhibitor cocktail | Cell Signaling Technology | #5871S | |
| Chemical compound, drug | Phospahtase inhibitor cocktail | Cell Signaling Technology | #5870S | |
| Software, algorithm | Graphpad | Prism | RRID:SCR_002798 | |
| Software, algorithm | Image Lab | Bio-Rad | RRID:SCR_014210 | |
| Software, algorithm | Aperio ImageScope | Leica | RRID:SCR_020993 | |
| Software, algorithm | SAMTOOLS | htslib.org | RRID:SCR_002105 | |
| Software, algorithm | Illumina Sequencing HUB (BaseSpace) | Illumina | RRID:SCR_011881 | |
| Software, algorithm | Cutadapt | code.google.com/p/cutadapt/ | RRID:SCR_011841 | |
| Software, algorithm | Bowtie 2 | bowtie-bio.sourceforge.net/ bowtie2/index.shtml | RRID:SCR_016368 | |
| Software, algorithm | ConsensusPathDB | Cpdb.molgen.mpg.de | RRID:SCR_002231 | |
| Software, algorithm | Reactome | https://reactome.org/ | RRID:SCR_003485 | |
| Software, algorithm | MAGeCK | https://sourceforge.net/projects/ mageck/ | | |
| Software, algorithm | STAR2.6.1a | https://code.google.com/archive/p/ rna-star/ | RRID:SCR_004463 | |
| Software, algorithm | DESeq2 | http://bioconducter.org/packages/ release/bioc/html/DESeq2/html | RRID:SCR_015687 | |
| Software, algorithm | CBioPortal | cbioportal.org | RRID:SCR_014555 | |
| Software, algorithm | The Cancer Genome Atlas (TCGA) | cancergenome.nih.gov | RRID:SCR_003193 | |
| Software, algorithm | R 3.6.2 | https://www.r-project.org/ | RRID:SCR_001905 | |
| Software, algorithm | BEAVR | https://github.com/ developerpiru/BEAVR; https://hub. docker.com/ r/pirunthan/beavr; *Perampalam and Dick, 2020* | | BMC Bioinformatics. 2020 May 29;21(1):221. |
| Other | Streptavidin HRP | Vector Laboratories | #SA-5704-100 | Secondary for IHCSee Methods, Histology and immunohisto-chemistry |
| Other | ImmPACT DAB | Vector Laboratories | #SK-4105 | Chromophore for IHCSee Methods, Histology and immunohisto-chemistry |
| Other | VectaMount | Vector Laboratories | #H-5700-60 | Mounting materialfor IHC glass slides See Methods, Histology and immunohisto-chemistry |
| Other | Hematoxylin | Millipore Sigma | #HHS32-1L | Counterstain reagentFor IHCSee Methods, Histology and immunohisto-chemistry |
| Other | Eosin Y | Millipore Sigma | #E4009 | Common stain in conjunction with hematoxylinfor IHC of tissue See Methods, Histologyand immunohisto-chemistry |

*Appendix 1 Continued on next page*

*Appendix 1 Continued*

| Reagent type (species) or resource | Designation | Source or reference | Identifiers | Additional information |
|---|---|---|---|---|
| Other | Gibson Assembly Mastermix | New England Biolabs | #E2611 | See "Methods" Generation of Knockout Lines |
| Other | Dynabeads | Thermo-Fisher | #10003D | See Methods, Dyrk1AIP kinase assay |
| Other | BsmB1 | New England Biolabs | #R0739 | Restriction endonucleasefor construction of CRISPR constructs |
| Other | EcoRV | New England Biolabs | #R0195 | Restriction endonuclease for construction of pcDNA NTN1. See "Methods"Generation of Overexpression cell lines |
| Other | XbaI | New England Biolabs | #R0145 | Restriction endonuclease for construction of pcDNA NTN1. See "Methods"Generation of Overexpression cell lines |
| Other | NotI | New England Biolabs | #R3189 | Restriction endonuclease for construction of pcDNA NTN3. See "Methods"Generation of Overexpression cell lines |
| Other | BamH1 | New England Biolabs | #R0136 | Restriction endonuclease for construction of FutdTWNTN1 and NTN3. See "Methods"Generation of Overexpression cell lines |
| Other | EcoRI | New England Biolabs | #R0101 | Restriction endonucleasefor construction of FutdTWNTN1 and NTN3. See "Methods"Generation of Overexpression cell lines |
| Sequence-based reagent | v2.1-F1 | TKO libraryProtocols Addgene#90294 | PCR Primer for genomic amplification of CRISPR guides http://dx.doi.org/10.1016/j.cell.2015.11.015 | 5'-GAGGGCCTATT TCCCATGATTC-3' |
| Sequence-based reagent | v2.1-R1 | TKO libraryProtocols Addgene#90294 | PCR Primer for genomic amplification of CRISPR guides http://dx.doi.org/10.1016/j.cell.2015.11.015 | 5'-GTTGCGAAAAA GAACGTTCACGG-3' |
| Sequence-based reagent | D501 -F | TKO libraryProtocols Addgene#90294 | Indexing Primer http://dx.doi.org/10.1016/j.cell.2015.11.015 | 5'-AATGATACGGCGACCACC GAGATCTACACTATAG CCTACACTCTTTCCCTACA CGACGCTCTTCCGAT CTTTGTGG AAAGGACGAAACACCG-3' |
| Sequence-based reagent | D502-F | TKO libraryProtocols Addgene#90294 | Indexing Primer http://dx.doi.org/10.1016/j.cell.2015.11.015 | 5'-AATGATACGGCGACCA CCGAGATCTACACATAGA GGCACACTCTTTCCCTA CACGACGCTCTTCCGAT CTTTGTGGA AAGGACGAAACACCG-3' |
| Sequence-based reagent | D503-F | TKO libraryProtocols Addgene#90294 | Indexing Primer http://dx.doi.org/10.1016/j.cell.2015.11.015 | 5'-AATGATACGGCGACCA CCGAGATCTACACCCT ATCCTACACTCTTTCCCT ACACGACGCTCTTCCG ATCTTTGTGGA AAGGACGAAACACCG-3' |
| Sequence-based reagent | D504-F | TKO libraryProtocols Addgene#90294 | Indexing Primer http://dx.doi.org/10.1016/j.cell.2015.11.015 | 5'-AATGATACGGCGACCACC GAGATCTACACGGCT CTGAACACTCTTTCCCTA CACGACGCTCTTCCG ATCTTTGTGG AAAGGACGAAACACCG-3' |
| Sequence-based reagent | D505-F | TKO libraryProtocols Addgene#90294 | Indexing Primer http://dx.doi.org/10.1016/j.cell.2015.11.015 | 5'-AATGATACGGCGACCA CCGAGATCTACACAGGC GAAGACACTCTTTCCCTA CACGACGCTCTTCCGAT CTTTGTGGA AAGGACGAAACACCG-3' |

*Appendix 1 Continued on next page*

*Appendix 1 Continued*

| Reagent type (species) or resource | Designation | Source or reference | Identifiers | Additional information |
|---|---|---|---|---|
| Sequence-based reagent | D506-F | TKO library Protocols Addgene#90294 | Indexing Primer http://dx.doi.org/10.1016/j.cell.2015.11.015 | 5'-AATGATACGGCGACCA CCGAGATCTACACTAA TCTTAACACTCTTTCCCTA CACGACGCTCTTCCG ATCTTTGTG GAAAGGACGAAACACCG-3' |
| Sequence-based reagent | D701-R | TKO library Protocols Addgene#90294 | Indexing Primer http://dx.doi.org/10.1016/j.cell.2015.11.015 | 5'-CAAGCAGAAGACGG CATACGAGATCGAGTAA TGTGACTGGAGTTCAGA CGTGTGCTCTTCCGAT CTACTTGCTAT TTCTAGCTCTAAAAC-3' |
| Sequence-based reagent | D702-R | TKO library Protocols Addgene#90294 | Indexing Primer http://dx.doi.org/10.1016/j.cell.2015.11.015 | 5'-AATGATACGGCGACCA CCGAGATCTACACATAGA GGCACACTCTTTCCCTA CACGACGCTCTTCCGAT CTTTGTGG AAAGGACGAAACACCG-3' |
| Sequence-based reagent | D704-R | TKO library Protocols Addgene#90294 | Indexing Primer http://dx.doi.org/10.1016/j.cell.2015.11.015 | 5'-CAAGCAGAAGACGGCA TACGAGATGGAATCTCGT GACTGGAGTTCAGACGT GTGCTCTTCCGATCTAC TTGCTATTTCTA GCTCTAAAAC-3' |
| Sequence-based reagent | D705-R | TKO libraryProtocols Addgene#90294 | Indexing Primer http://dx.doi.org/10.1016/j.cell.2015.11.015 | 5'-CAAGCAGAAGACGGCAT ACGAGATTTCTGAATGTG ACTGGAGTTCAGACGTGTG CTCTTCCGATCTACTTG CTATTTCTA GCTCTAAAAC-3' |
| Sequence-based reagent | D706-R | TKO libraryProtocols Addgene#90294 | Indexing Primer http://dx.doi.org/10.1016/j.cell.2015.11.015 | 5'-CAAGCAGAAGACGGCAT ACGAGATACGAATTCGT GACTGGAGTTCAGACGTGTGCTCT TCCGATCTAC TTGCTATTTCTAGCTCTAAAAC-3' |
| Sequence-based reagent | D707-R | TKO libraryProtocols Addgene#90294 | Indexing Primer http://dx.doi.org/10.1016/j.cell.2015.11.015 | 5'-CAAGCAGAAGACGG CATACGAGATAGCTTCAGGT GACTGGAGTTCAGAC GTGTGCTCTTCCGATCTAC TTGCTATTTCTAG CTCTAAAAC-3' |
| Sequence-based reagent | Dyrk1A sgRNA-A | This paper | CRISPR guide | 5'-CTCACTTAT CTTCTTGTAGG-3' |
| Sequence-based reagent | Dyrk1A sgRNA-B | This paper | CRISPR guide | 5'-GCAACGTG GGATTATGGATT-3' |
| Sequence-based reagent | NTN1-BsgRNA | GeCKO genomicCRISPRLibrary V2Addgene#1000000049 | CRISPR guide | 5'-ACCCGTCAC GCCGTCCTTGC-3' |
| Sequence-based reagent | NTN1-CsgRNA | GeCKO genomicCRISPRLibrary V2Addgene#1000000049 | CRISPR guide | 5'-TATCGGCCA CGATGCCGCTC-3' |
| Sequence-based reagent | NTN3-BsgRNA | GeCKO genomicCRISPRLibrary V2Addgene#1000000049 | CRISPR guide | 5'-GGTCTCGAT AGAAGCCCTCC-3' |
| Sequence-based reagent | NTN3-CsgRNA | GeCKO genomicCRISPRLibrary V2Addgene#1000000049 | CRISPR guide | 5'-ACCTGCAAC CGCTGCGCGCC-3' |
| Sequence-based reagent | NTN4-BsgRNA | GeCKO genomicCRISPRLibrary V2Addgene#1000000049 | CRISPR guide | 5'-CGCAGGTC ACGATAGAAGCC-3' |
| Sequence-based reagent | NTN5-BsgRNA | GeCKO genomicCRISPRLibrary V2Addgene#1000000049 | CRISPR guide | 5'-GCCGCCCG TCCCATCGAGAC-3' |
| Sequence-based reagent | NTN5-CsgRNA | GeCKO genomicCRISPRLibrary V2Addgene#1000000049 | CRISPR guide | 5'-GGCCTGAC CTGCAACCGCTG-3' |

*Appendix 1 Continued on next page*

*Appendix 1 Continued*

| Reagent type (species) or resource | Designation | Source or reference | Identifiers | Additional information |
|---|---|---|---|---|
| Sequence-based reagent | Unc5A-2sgRNA | GeCKO genomicCRISPRLibrary V2Addgene#1000000049 | CRISPR guide | 5'-CGCCCG CGGCCATGGCCGTC-3' |
| Sequence-based reagent | Unc5A-3sgRNA | GeCKO genomicCRISPRLibrary V2Addgene#1000000049 | CRISPR guide | 5'-GTCCTCGCCG CTTGGCTCCG-3' |
| Sequence-based reagent | Unc5B-2sgRNA | GeCKO genomicCRISPRLibrary V2Addgene#1000000049 | CRISPR guide | 5'-TTCACAAT GTAGGCGTCCTG-3' |
| Sequence-based reagent | Unc5B-3sgRNA | GeCKO genomicCRISPRLibrary V2Addgene#1000000049 | CRISPR guide | 5'-CCTGTGTGA CGTGGTCGTTC-3' |
| Sequence-based reagent | Unc5C-2sgRNA | GeCKO genomicCRISPRLibrary V2Addgene#1000000049 | CRISPR guide | 5'-TGAGATTT CGCGCCAGCAAG-3' |
| Sequence-based reagent | Unc5C-3sgRNA | GeCKO genomicCRISPRLibrary V2Addgene#1000000049 | CRISPR guide | 5'-CAATGCGC ACATACGCCTTC-3' |
| Sequence-based reagent | Unc5D-2sgRNA | GeCKO genomicCRISPRLibrary V2Addgene#1000000049 | CRISPR guide | 5'-GCGCTTA CCTCGGGCAGCCG-3' |
| Sequence-based reagent | Unc5D-3sgRNA | GeCKO genomicCRISPRLibrary V2Addgene#1000000049 | CRISPR guide | 5'-CAGAGACGT GCTCGTTCTGA-3' |
| Sequence-based reagent | DCC-2sgRNA | GeCKO genomicCRISPRLibrary V2Addgene#1000000049 | CRISPR guide | 5'-AAATTCCAA TGTCCCCCGGT-3' |
| Sequence-based reagent | DCC-3sgRNA | GeCKO genomicCRISPRLibrary V2Addgene#1000000049 | CRISPR guide | 5'-GCAGATCA GCCGACTCCAAC-3' |
| Sequence-based reagent | Neo1-2sgRNA | GeCKO genomicCRISPRLibrary V2Addgene#1000000049 | CRISPR guide | 5'-GAACCTTC CTCAGTTTATGC-3' |
| Sequence-based reagent | Neo1-3sgRNA | GeCKO genomicCRISPRLibrary V2Addgene#1000000049 | CRISPR guide | 5'-TGTTTCCCA CGTAACAGTGA-3' |
| Sequence-based reagent | DSCAM-2sgRNA | GeCKO genomicCRISPRLibrary V2Addgene#1000000049 | CRISPR guide | 5'-AGTGATGTAC GCCTCCACCG-3' |
| Sequence-based reagent | DSCAM-3sgRNA | GeCKO genomicCRISPRLibrary V2Addgene#1000000049 | CRISPR guide | 5'-GGAGCCCTAT ACAGTCCGTG-3' |
| Sequence-based reagent | Dyrk1A Exon2 F | | PCR prime | 5'-GGTTTCACCT GGTTTGGGGA-3' |
| Sequence-based reagent | Dyrk1A Exon2 R | | PCR prime | 5'-TCCGTGGG CAAGAAACTTT-3' |
| Sequence-based reagent | Unc5A-FqRCR primer | | PCR primer | 5'-GCTGAGGCGC TAAAGCCGCCCTC-3' |
| Sequence-based reagent | Unc5A-RqRCR primer | | PCR primer | 5'-ACCTGCTGCCT TGAGACATTAATGC-3' |
| Sequence-based reagent | Unc5B-FqPCR primer | | PCR primer | 5'-CCCGCCACA CAGATCTACTT-3' |
| Sequence-based reagent | Unc5B-RqPCR primer | | PCR primer | 5'-CAGTAATCC TCCAGCCCAAA-3' |
| Sequence-based reagent | Unc5C_1-FqPCR primer | | PCR primer | 5'-GCAAATTGCTG GCTAAATATCAGGAA-3' |
| Sequence-based reagent | Unc5C_1-RqPCR primer | | PCR primer | 5'-GCTCCACTGTGT TCAGGCTAAATCTT-3' |
| Sequence-based reagent | Unc5C_2-FqPCR primer | | PCR primer | 5'-AATTGATCC CGTTGAAGATCGG-3' |

*Appendix 1 Continued on next page*

*Appendix 1 Continued*

| Reagent type (species) or resource | Designation | Source or reference | Identifiers | Additional information |
|---|---|---|---|---|
| Sequence-based reagent | Unc5C_2-RqPCR primer | | PCR primer | 5′-TGACAGTGG CAGTTGTACTTTT-3′ |
| Sequence-based reagent | Unc5D_1-FqPCR primer | | PCR primer | 5′-CAAGAGCAA CCCTATTGCACT-3′ |
| Sequence-based reagent | Unc5D_1-RqPCR primer | | PCR primer | 5′-CTCGTTCTG ATGGACCCACT-3′ |
| Sequence-based reagent | Unc5D_2-FqPCR primer | | PCR primer | 5′-CAAGAGCAA CCCTATTGCACT-3′ |
| Sequence-based reagent | Unc5D_2_RqPCR primer | | PCR primer | 5′-AAGCCCTTCC CGAATCCATC-3′ |
| Sequence-based reagent | NTN1-FqPCR primer | | PCR primer | 5′-TGCAAGAAGG ACTATGCCGTC-3′ |
| Sequence-based reagent | NTN1-RqPCR primer | | PCR primer | 5′-GCTCGTGCCC TGCTTATACAC-3′ |
| Sequence-based reagent | NTN3-FqPCR primer | | PCR primer | 5′-TGCAAGCCCT TCTACTGCGACA-3′ |
| Sequence-based reagent | NTN3-RqPCR primer | | PCR primer | 5′-CAGTCGGTA CAGCTCCATGTTG-3′ |
| Sequence-based reagent | NTN4-FqPCR primer | | PCR primer | 5'-CAGAAGGACAG TATTGCCAGAGG-3' |
| Sequence-based reagent | NTN4-RqPCR primer | | PCR primer | 5'-GCAGAAGGTC ACTGAGTTGGCA-5' |
| Sequence-based reagent | NTN5-FqPCRprimer | | PCR primer | 5'-CTTGCCACTA CTCCTGGTGCTT-3' |
| Sequence-based reagent | NTN5-RqPCR primer | | PCR primer | 5'-AGTACCTC CGAAGGCTCATGTG-3' |
| Sequence-based reagent | DCC_1-FqPCR primer | | PCR primer | 5′-GACTTTACCAAT GTGAGGCATCT-3′ |
| Sequence-based reagent | DCC_1-RqPCR primer | | PCR primer | 5′-GGTCCTGCT ACTGCAACTTTT-3′ |
| Sequence-based reagent | DCC_2-FqPCRprimer | | PCR primer | 5′-GAGACACA GTGCTACTCAAGTG-3′ |
| Sequence-based reagent | DCC_2-RqPCR primer | | PCR primer | 5′-GGAGTCAGG TCTTGTTGGTTCTT-3′ |
| Sequence-based reagent | DSCAM_1-FqPCR primer | | PCR primer | 5′-TTGCGGTCT TCAAGTGCATTA-3′ |
| Sequence-based reagent | DSCAM_1-RqPCR primer | | PCR primer | 5′-TGCAGCGGTAGTTATACAATCCA-3′ |
| Sequence-based reagent | DSCAM_2-FqPCR primer | | PCR primer | 5′-ATCAGACCCAGCGAACTCAG-5′ |
| Sequence-based reagent | DSCAM_2-RqPCR primer | | PCR primer | 5′-CCAGCGGTAATCTGGCTCAG-3′ |
| Sequence-based reagent | Neo-1-FqPCR primer | | PCR primer | 5′-GTCACTGAGACCTTGGTAAGCG-3′ |
| Sequence-based reagent | Neo-1-RqPCR primer | | PCR primer | 5′-TCAGCAGACAGCCAGTCAGTTG-3′ |
| Sequence-based reagent | GAPDH-FqPCR primer | | PCR primer | 5'-CGGAGTCAACGGATTTGGTCGTAT-3' |
| Sequence-based reagent | GAPDH-RqPCR primer | | PCR primer | 5'-AGCCTTCTCCATGGTGGTGAAGAC-3' |
| Sequence-based reagent | NTN1-F cloning primer_1 | | Cloning primer | 5′- GCTATCGATATCCCAAACG CCACCATGATGCGCGC TGTGTGGGAGGCGCTG -3′ |
| Sequence-based reagent | NTN1-Rcloning primer | | Cloning primer | 5′- GCTATCCTCGAGGGCCTTCTTGCAC TTGCCCTTCTTCTCCCG -3′ |

*Appendix 1 Continued on next page*

*Appendix 1 Continued*

| Reagent type (species) or resource | Designation | Source or reference | Identifiers | Additional information |
|---|---|---|---|---|
| Sequence-based reagent | NTN3-Fcloning primer_1 | | Cloning primer | 5'-GCTAGCGCGGCCGCCACCATGC CTGGCTGGCCCTGG-3' |
| Sequence-based reagent | NTN3-Rcloning primer | | Cloning primer | 5'- ACGCGTGAATTCTTATCAA CCGGTATGCATATTCA GATCCTCTTCTGAGAT -3' |
| Sequence-based reagent | NTN1-Fcloning primer_2 | | Cloning primer | 5'-CACTGTAGATCTCCAAACGC CACCATGATGCGCGCTG TGTGGGAGGCGCTG-3' |
| Sequence-based reagent | NTN3-FCloning primer_2 | | Cloning primer | 5'-CTCGAGATCTGCGGCCGCCACC ATGCCTGGCTGGCCCTGG-3' |
| Sequence-based reagent | NTN1_NTN3Cloning reverse primer | | Cloning primer | 5'-ACGCGTGAATTCTTATCAA CCGGTATGCATATTCA GATCCTCTTCTGAGAT-3' |
| Sequence-based reagent | LacZ control CRISPR guide | | CRISPR guide | 5'-CCCGAATCTCTA TCGTGCGG-3' |
| Sequence-based reagent | PSMD1-1 | | Fitness gene CRISPR guide http://dx.doi.org/10.1016/ j.cell.2015.11.015 | 5'-TGTGCGCTACG GAGCTGCAA-3' |
| Sequence-based reagent | PSMD1-5 | | Fitness gene CRISPR guide http://dx.doi.org/10.1016/ j.cell.2015.11.015 | 5'-ACCAGAGCCAC AATAAGCCA-3' |
| Sequence-based reagent | EIF3D | | Fitness gene CRISPR guide http://dx.doi.org/10.1016/ j.cell.2015.11.015 | 5'-ACCGACTTCAAC TGAAGAGTCTCG-3' |
| Sequence-based reagent | PSMB2 | | Fitness gene CRISPR guide http://dx.doi.org/10.1016/ j.cell.2015.11.015 | 5'-AATATTGTCCAGA TGAAGGA-3' |

