## [Editor Report · eLife assessment]

The authors further corroborated their model that Netrin signaling promotes survival and dissemination of non-proliferating ovarian cancer cells. These **valuable** results were found to be of significant potential interest to cancer biologists in as much as they address gaps in knowledge pertinent to the mechanisms underpinning ovarian cancer spread. In general, it was thought that **solid** experimental evidence was provided to support the role of Netrin signaling in fueling ovarian cancer progression.

---

## [Referee Report · Joint public review]

In this article, the authors employed modified CRISPR screens ["guide-only (GO)-CRISPR"] in the attempt to identify the genes which may mediate cancer cell dormancy in the high grade serous ovarian cancer (HGSOC) spheroid culture models. Using this approach, they observed that abrogation of several of the components of the netrin (e.g., DCC, UNC5Hs) and MAPK pathways compromise survival of non-proliferative ovarian cancer cells. This strategy was complemented by the RNAseq approach which revealed that number of the components of the netrin pathway are upregulated in non-proliferative ovarian cancer cells, and that their overexpression is lost upon disruption of DYRK1A kinase that has been previously demonstrated to play a major role in survival of these cells. Perampalam et al. then employed a battery of cell biology approaches to support the model whereby the Netrin signaling governs the MEK-ERK axis to support survival of non-proliferative ovarian cancer cells. Moreover, the authors show that overexpression of Netrins 1 and 3 bolsters dissemination of ovarian cancer cells in the xenograft mouse model, while also providing evidence that high levels of the aforementioned factors are associated with poor prognosis of HGSOC patients.

Strengths:

In this valuable study Perampalam et al. developed a CRISPR-based screening approach to identify key genes that are enriched in high grade serous ovarian cancer spheroids. This led to a discovery that Netrin signaling plays a prominent role in survival of ovarian cancer cells. During revision, the authors provide additional evidence to support their central claims and to this end, it was found that they now provide solid evidence to substantiate the proposed model. This work is anticipated to be of interest to cancer biologists specializing in ovarian cancer biology.

---

## [Author Response]

The following is the authors’ response to the original reviews.

**Reviewer #1 (Public Review):**
Summary:Perampalam et al. describe novel methods for genome-wide CRISPR screening to identify and validate genes essential for HGSOC spheroid viability. In this study, they report that Netrin signaling is essential for maintaining disseminated cancer spheroid survival, wherein overexpression of Netrin pathway genes increases tumor burden in a xenograft model of ovarian cancer. They also show that high netrin expression correlates with poor survival outcomes in ovarian cancer patients. The study provides insights into the biology of netrin signaling in DTC cluster survival and warrants development of therapies to block netrin signaling for treating serous ovarian cancer.Strengths:- The study identifies Netrin signaling to be important in disseminated cancer spheroid survival- A Novel GO-CRISPR methodology was used to find key genes and pathways essential for disseminated cancer cell survival

Thanks for the endorsement of our work and its importance to metastasis in ovarian cancer.

Weaknesses:- The term dormancy is not fully validated and requires additional confirmation to claim the importance of Netrin signaling in "dormant" cancer survival.- Findings shown in the study largely relate to cancer dissemination and DTS survival rather than cancer dormancy.

Much of the validation of dormancy and cell cycle arrest in HGSOC spheroids, as well as the culture model, have been published previously and hence was not repeated here. I think this reviewer will appreciate the updated citations and explanations to better illustrate the state of knowledge. We have also added new experiments that further emphasize the dormant state of spheroid cells in culture and xenografts, as well as patient derived spheroids used in this study.

**Reviewer #1 (Recommendations for Authors):**
(1) It is unclear what spheroid/adherent enrichment ratio is and how it ties into genes affecting cell viability. Why is an ER below 1 the criteria for selecting survival genes?

Our screen uses the ‘guide only’ comparison in each culture condition to establish a gene score under that specific condition. A low adherent score captures genes that are essential under standard culture conditions where cells are proliferating and this can include genes needed for proliferation or other basic functions in cell physiology. A low spheroid score identifies the genes that are most depleted in suspension when cells are growth arrested and this is an indication of cell death in this condition. Since gene knock outs are first established in adherent proliferating conditions, essential genes under these conditions will already start to become depleted from the population before suspension culture. By selecting genes with a ratio of <1 we can identify those that are most relevant to dormant suspension culture conditions. Ultimately, the lowest enrichment ratio scores represent genes whose loss of function is dispensable in the initial adherent condition, but critical for survival in suspension and this is what we aimed to identify. We’ve updated Figure 1B to illustrate this and we’ve updated the explanation of the enrichment ratio on page 6, lines 144 to 147 of the results.

(2) The WB for phospho-p38 in figure 1A for OVCAR8 line does not show increased phosphorylation in the spheroid relative to the adherent. If anything, phospho-p38 appears to be reduced in the spheroid. Can the authors provide a better western blot?

We’ve updated this blot with a longer exposure, see Figure 1A. Phosphorylation levels of p38 are essentially unchanged in OVCAR8 cells in suspension culture, although the overall levels of p38 may be slightly reduced in dormant culture conditions.

(3) How did the authors confirm dormancy apart from western blot for phospho-ERK vs phospho-p38? Authors should add EdU/BrdU staining and/or Ki67 staining to confirm dormancy.

Previous publications that appear as citations 7,10, and 33 in the reference list established the growth arrest state of these cells in suspension culture in the past. This included measuring other known markers of dormancy and quiescence such as p27, p130, and reduced cyclin/cdk activity and 3H-thymidine incorporation. In addition, other associated characteristics of dormancy such as EMT and catabolic metabolism have been demonstrated in these culture conditions (see citation 11 and Rafehi et al. Endocr. Relat. Cancer 23;147-59). We’ve added these additional citations to our descriptions of dormant spheroid culture to better clarify the status of these cells in our experiments (see page 6, lines 126-28). To ensure that cells are growth arrested in the experiments shown in this paper, we have updated Figure 1A to include blots of p130 and Ki67 to further emphasize that spheroid cells are not proliferating as the quiescence marker (p130) is high and the proliferative marker (Ki67) is lost in suspension culture.

(4) Can the authors report spheroid volume over time in culture? How was viability measured?

We’ve updated the methods (see page 27, line 574) to better highlight the description of cell survival that answers both of these questions. At the ends of experimental time points in both the screen and viability assays we captured live cells by replating on adherent plasticware. We fixed and stained with crystal violet and photographed plates to illustrate the sizes of spheroids (shown in Fig. 2 Supplement 1E, Fig. 6C, and 7D). We subsequently extracted the dye and quantitated it spectrophotometrically to quantitatively compare biomass of viable cells between experiments irrespective of the relatively random shapes of spheroids. We found reattachment and staining in this manner to match traditional viability assays such as CellTiter-Glo in a previous paper (10). Furthermore, biomass never increases in culture and diminishes gradually over time in culture consistent with the non-proliferative state of these experiments. Double checks of this equivalency of viability and reattached biomass measurments, as well as demonstrating that biomass is lost over time, are shown in Fig. 2 Supplement 1E that compares reattached crystal violet staining measurements with CellTiter-Glo for DYRK1A knock out cells over time in culture. In addition, we include a comparison of crystal violet staining of reattached spheroids with trypan blue dye exclusion in Fig. 5G and H. In both cases reattachment and more direct viability assays demonstrate the same conclusion that Netrin signaling supports viability in dormant culture.

(5) Please show survival significance of Netrin signaling genes in recurrence/relapse free survival to claim importance in cancer dormancy.

See Fig. 7 Supplement 1C where we include the recurrence free survival data. Netrin-1, and -3 high expressors also have a numerically shorter progression free survival but it is not statistically significant. Netrin-1 overexpression alone is also shown and it shows shorter survival with a P-value of 0.0735. Elevated survival of dormant cells in a residual disease state is expected to increase the chance of relapse and shorten this interval. Thus, this data is consistent with our model, but lacks statistical significance.

There are many alternative ways to interpret what shorter progression free survival, or overall survival, may mean biologically. Since survival of dormant cells is but one of them, we also added new data to experimentally investigate the role of endogenous Netrin signaling in dormant residual disease in Fig. 6 and described on page 12, lines 266-87. We used xenograft experiments to show OVCAR8 spheroids form and withdraw from the cell cycle equivalently to suspension culture following intraperitoneal injection. Furthermore, loss of Netrin signaling due to receptor deletions compromises survival during this early window before disseminated lesions form. This argues that Netrin signaling contributes to survival during this window of dormancy. In addition, mice engrafted with mutant cells experience prolonged survival when Netrin signaling is blocked. Together, these experiments further argue that Netrin signaling supports survival in the dormant, non-proliferative phase, and leads to reduced survival of mice.

(6) The authors show IHC staining of patient ascites derived HGSOC spheroids. However, no marker for dormancy is shown in these spheroids. Adding Ki67 staining or phospho-ERK vs phospho-p38 would be necessary to confirm cancer dormancy.

We have added new staining for Ki67 and p130 that compares these markers in HGSOC tumors where Ki67 is high and p130 is low with ascites derived spheroids where staining is the opposite. Importantly, expression of p130 is linked to cellular quiescence and is not found to accumulate in the nucleus of cells that are just transiting through G1. This confirms that the ascites derived spheroids are dormant. See Fig. 4A-E and described on page 9, lines 201-7.

(7) Overall, the findings are interesting in the context of cancer dissemination. There is not enough evidence for cancer dormancy and the importance of Netrin signaling in the survival of cancer dormancy. Overexpression of Netrin increases phosphorylation of ERK, leading one to expect an increase in proliferation. This suggests that Netrin breaks cancer cells out of dormancy, into a proliferative state.

We have found that the discovery of Netrin activation of MEK-ERK in growth arrested cells is counterintuitive to many cancer researchers. However, this axis exists in other paradigms of Netrin signaling in axon outgrowth that are not proliferation related (see citation 26, Forcet et al. Nature 417; 443-7 as an example). We have added Fig. 5D and descriptions on page 11, lines 244-52 to better clarify that Netrins CAN’T induce cell proliferation through ERK. Addition of recombinant Netrin-1 can only induce ERK phosphorylation in suspension culture conditions and not in quiescent adherent conditions. The small magnitude of ERK phosphorylation induced by Netrin-1 in suspension compared to treating adherent, quiescent cells with the same concentration of mitogenic EGF further emphasizes that this is not a proliferative signal. Lastly, the new xenograft experiment in Fig. 6A-D described on page 12, lines 266-81 demonstrates the growth arrested context in which Netrin signaling in dormant spheroids leads supports viability.

(8) If authors wish to claim cancer dormancy as the premise of their study, additional confirmatory experiments are required to support their claims. Alternatively, based on the current findings of the study, it would be best to change the premise of the article to Netrin signaling in cancer dissemination and survival of disseminated cancer spheroids rather than cancer dormancy.

I expect that this reviewer will agree that we have added more than sufficient explanations of background work on HGSOC spheroid dormancy from the literature, as well as new experiments that address their questions about dormancy in our experiments.

**Reviewer #2 (Public Review):**

**Summary:**
In this article, the authors employed modified CRISPR screens ["guide-only (GO)-CRISPR"] in the attempt to identify the genes which may mediate cancer cell dormancy in the high grade serous ovarian cancer (HGSOC) spheroid culture models. Using this approach, they observed that abrogation of several of the components of the netrin (e.g., DCC, UNC5Hs) and MAPK pathways compromise the survival of non-proliferative ovarian cancer cells. This strategy was complemented by the RNAseq approach which revealed that a number of the components of the netrin pathway are upregulated in non-proliferative ovarian cancer cells and that their overexpression is lost upon disruption of DYRK1A kinase that has been previously demonstrated to play a major role in survival of these cells. Perampalam et al. then employed a battery of cell biology approaches to support the model whereby the Netrin signaling governs the MEK-ERK axis to support survival of non-proliferative ovarian cancer cells. Moreover, the authors show that overexpression of Netrins 1 and 3 bolsters dissemination of ovarian cancer cells in the xenograft mouse model, while also providing evidence that high levels of the aforementioned factors are associated with poor prognosis of HGSOC patients.Strengths:Overall it was thought that this study is of potentially broad interest in as much as it provides previously unappreciated insights into the potential molecular underpinnings of cancer cell dormancy, which has been associated with therapy resistance, disease dissemination, and relapse as well as poor prognosis. Notwithstanding the potential limitations of cellular models in mimicking cancer cell dormancy, it was thought that the authors provided sufficient support for their model that netrin signaling drives survival of non-proliferating ovarian cancer cells and their dissemination. Collectively, it was thought that these findings hold a promise to significantly contribute to the understanding of the molecular mechanisms of cancer cell dormancy and in the long term may provide a molecular basis to address this emerging major issue in the clinical practice.

Thanks for the kind words about the importance of our work in the broader challenges of cancer treatment.

Weaknesses:Several issues were observed regarding methodology and data interpretation. The major concerns were related to the reliability of modelling cancer cell dormancy. To this end, it was relatively hard to appreciate how the employed spheroid model allows to distinguish between dormant and e.g., quiescent or even senescent cells. This was in contrast to solid evidence that netrin signaling stimulates abdominal dissemination of ovarian cancer cells in the mouse xenograft and their survival in organoid culture. Moreover, the role of ERK in mediating the effects of netrin signaling in the context of the survival of non-proliferative ovarian cancer cells was found to be somewhat underdeveloped.

Experiments previously published in citation 7 show that growth arrest in patient ascites derived spheroids is fully reversible and that argued against non-proliferative spheroids being a form of senescence and moved this work into the dormancy field. We have added extensive new support for our model systems and data to address the counterintuitive aspects of MEK-ERK signaling in survival instead of proliferation.

**Reviewer #1 Recommendations for Authors**
(1) A better characterization of the spheroid model may be warranted, including staining for the markers of quiescence and senescence (including combining these markers with staining for the components of the netrin pathway)

See Figure 1A and page 6, lines 126-36 where we have added blots for Ki67 and p130 to better emphasize the arrested proliferative state of cells in our screening conditions. We have also added these same controls for patient ascites-derived spheroids in Figure 4 and described on page 9, lines 203-7. One realization from this CRISPR screen, and others in our lab, is that it identifies functionally important aspects of cell physiology and not necessarily ones that are easily explored using commercially available antibodies. Netrin-1 and -3 staining of patient derived spheroids in Fig. 4, as well as cell line spheroids stained in Fig. 4 Supplement 1 further support the relevance of this pathway in dormant cancer cells because Netrins are expressed in the right place at the right time. The Netrin-1 stimulation experiments in Fig. 5C were originally carried out to probe HGSOC cells for functionality of Netrin receptors since we couldn’t reliably detected them by blotting or staining with available antibodies. This demonstrates that this pathway is active in the various HGSOC cell lines we’ve used and specifically, using OVCAR8 cells, we show it is only active in suspension culture conditions.

(2) In figure 1A it appears that total p38 levels are reduced in some cell lines in spheroid vs. adherent culture. The authors should comment on this.

These blots have been updated to be more clear. Overall p38 levels may be reduced in some cell lines and when compared with activation levels of phosphorylated p38 it suggests the fraction of activated p38 is higher. OVCAR8 cells may be an exception where the overall activity level remains approximately the same.

(3) The authors should perhaps provide a clearer rationale for choosing to focus on the netrin signaling vs. e.g., GPCR signaling, and consider more explicit defining of "primary" vs. "tertiary" categories in Reactome gene set analysis.

We’ve updated Fig. 1E and the text on page7, lines 161-5 to illustrate which gene categories identified in the screen belong to which tiers of Reactome categories. It better visualizes why we have investigated the Axon guidance pathway that includes Netrin because it is a highly specific signaling pathway that scores similarly to the broader and less specific categories at the very top of the list. As an aside, the GPCR signaling and GPCR downstream signaling have proven to be fairly intractable categories. As best we can tell the GPCR downstream signaling category is full of MAPK family members and likely represents some redundancy with MAPK further down.

(4) In figure 3A-C, including factors whose expression did not appear to change between adherent and suspension conditions may be warranted as the internal control. Figure 3D-F may benefit from some sort of quantification.

The mRNA expression levels are normalized to GAPDH as an internal control. We have updated this figure and re-plotted it as fold change relative to adherent culture cells with statistical comparisons to indicate which are significantly upregulated in suspension culture.

The IHC experiments are now in Fig. 4D-F and show positive staining for Netrin-1 and -3. Netrin-3 is easiest to see, while Netrin-1 is trickier because the difference with the no primary antibody control isn’t intensity, but the tint of the DAB stain. We had to counter stain the patient spheroids with Hematoxylin in order for the slide scanner to find the best focal plane and make image registration between sections possible. This unfortunately makes the Netrin-1 staining rather subtle. For cell line spheroids in the Fig. 4, Supplement 1 we didn’t need the slide scanner and show negative controls without counter stain that are much more convincing of Netrin-1 detection and reassure us that our staining detects the intended target. We’ve updated the labels in Fig. 4 and Fig. 4, Supplement 1 for this to be more intuitive. Unfortunately, relying on the tint of the DAB stain leaves this as a qualitative experiment.

- In figure 4C-E the authors show that Netrin-1 stimulation induces ERK phosphorylation whereby it is argued that this is a "low-level" stimulation of ERK signaling required for the survival of ovarian cells in the suspension. This is however hard to appreciate, and it was thought that having adherent cells in parallel would be helpful to wage whether this indeed is a "low level" ERK activity. Moreover, the authors should likely include downstream substrates of ERK (e.g., RSKs) as well as p38 in these experiments. The control experiments for the effects of PD184352 on ERK phosphorylation also appear to be warranted. Finally, performing the experiments with PD184352 in the presence of Netrin-1 stimulation would also be advantageous.

We have added a new Netrin-1 stimulation experiment in Fig. 4D (described on page 11, line 244-52) that shows that Netrins can only activate very low levels of ERK phosphorylation in suspension when proliferation is arrested. Netrin-1 stimulation of quiescent adherent cells where stimulation of proliferation is possible shows that Netrins are unable to activate ERK phosphorylation in this condition. In contrast, we also stimulate quiescent adherent OVCAR8 cells with an equal concentration of EGF (a known mitogen) to offer high level ERK phosphorylation as a side by side comparison. I think that this offers clear evidence that Netrin signaling is inconsistent with inducing cell proliferation. We’ve also updated citations in the introduction to include citation 26 that offers a previously reported paradigm of Netrin-ERK signaling in axon outgrowth that is a non-cancer, non-proliferative context to remind readers that Netrins utilize MEK-ERK differently.

We highlight Netrin-MEK-ERK signaling as key to survival for a number of reasons. First, Netrin signaling in this paradigm does not fit the dependence receptor paradigm where loss of Netrin receptors protect against cell death. Fig. 5B rules this out as receptor loss never offers a survival advantage, but clearly receptor deletions compromise survival in suspension culture. Second, positive Netrin signaling is known to support survival by inactivating phosphorylation of DAPK1. We’ve added this experiment as Fig. 5 Supplement 1D and show that loss of Netrin receptors doesn’t reduce DAPK1 phosphorylation in a time course of suspension culture. Consequently, we conclude this isn’t the survival signal either. Since MEK and ERK family members scored in our screen, we investigated their role in survival. We now show two different MEK inhibitors with different inhibitory mechanisms to confirm that MEK inhibition induces cell death. In addition to the previous PD184352 inhibitor in our first submission, we’ve added Trametinib as well and this is shown in Fig. 5G. Since it is surprising the MEK inhibition can kill instead of just arrest proliferation, we’ve also added another cell death assay in which we show trypan blue dye exclusion as a second look at survival. This is now Fig. 5H. Lastly, we include Trametinib inhibition of ERK phosphorylation in these assays in Fig. 5I. While we leave open what takes place downstream of ERK, our model in Fig. 5J offers a very detailed look at the components upstream.

- Does inhibition of ERK prevent the abdominal spread of ovarian cancer cells? The authors may feel that this is out of the scope of the study, which I would agree with, but then the claims regarding ERK being the major mediator of the effects of netrin signaling should be perhaps slightly toned down.

We agree that loss of function xenograft experiments will enhance our discovery of Netrin’s role in dormancy and metastasis. We have added a new Fig. 6 that uses xenografts with Netrin receptor deficient OVCAR8 cells (UNC5 4KO). It demonstrates that two weeks following IP engraftment we can isolate spheroids from abdominal washes and that cells have entered a state of reduced proliferation as determined by lowered Ki67 expression as well as other proliferation inducing genes. In the case of UNC5 4KO cells, there is significant attrition of these cells as determined by recovering spheroids in adherent culture (Fig.6C) and by Alu PCR to detect human cells in abdominal washes (Fig. 6D). Lastly, xenografts of UNC5 4KO cells cause much less aggressive disease and significantly extend survival of these mice (Fig. 6E,F). Not exactly the experiment that the reviewer is asking for, but a clear indication that Netrin signaling supports survival in xenograft model of dormancy.

- Notwithstanding that this could be deduced from figures 6D and F, it would be helpful if the number of mice used in each experimental group is clearly annotated in the corresponding figure legends. Moreover, indicating the precise statistical tests that were used in the figures would be helpful (e.g., specifying whether anova is one-way, two-way, or?)

We have added labels to what is now Fig. 8B to indicate the number of animals used for each genotype of cells. We have also updated figure legends to include more details of statistical tests used in each instance.